# PAC PREDICTION SETS UNDER LABEL SHIFT

**Wenwen Si**[1]**, Sangdon Park**[2]**, Insup Lee**[1]**, Edgar Dobriban**[3]**, Osbert Bastani**[1]

[1]Department of Computer & Information Science, University of Pennsylvania
[2]Graduate School of AI and Computer Science & Engineering, POSTECH
[3]Department of Statistics and Data Science, University of Pennsylvania
{wenwens, lee, obastani}@seas.upenn.edu, sangdon@postech.ac.kr
dobriban@wharton.upenn.edu

## ABSTRACT

Prediction sets capture uncertainty by predicting sets of labels rather than individual labels, enabling downstream decisions to conservatively account for all plausible outcomes. Conformal inference algorithms construct prediction sets guaranteed to contain the true label with high probability. These guarantees fail to hold in the face of distribution shift, which is precisely when reliable uncertainty quantification can be most useful. We propose a novel algorithm for constructing prediction sets with PAC guarantees in the label shift setting, where the probabilities of labels can differ between the source and target distributions. Our algorithm relies on constructing confidence intervals for importance weights by propagating uncertainty through a Gaussian elimination algorithm. We evaluate our approach on four datasets: the CIFAR-10 and ChestX-Ray image datasets, the tabular CDC Heart Dataset, and the AGNews text dataset. Our algorithm satisfies the PAC guarantee while producing smaller prediction set sizes compared to several baselines.

## 1 INTRODUCTION

Uncertainty quantification can be a critical tool for building reliable systems from machine learning components. For example, a medical decision support system can convey uncertainty to a doctor, or a robot can act conservatively with respect to uncertainty. These approaches are particularly important when the data distribution shifts as the predictive system is deployed, since they enable the decision-maker to react to degraded performance.

Conformal prediction (Vovk et al., 2005; Angelopoulos & Bates, 2021) is a promising approach to uncertainty quantification, aiming to outputs sets of labels instead of a single label. Under standard assumptions (i.i.d. or exchangeable data), it guarantees that the prediction set contains the true label with high probability. We consider *probably approximately correct (PAC)* (or *calibration-set-conditional*) guarantees (Vovk, 2012; Park et al., 2019), which ensure high probability coverage over calibration datasets used to construct the prediction sets.

In this paper, we propose a novel prediction set algorithm that provides PAC guarantees under the *label shift* setting, where the distribution of the labels may shift, but the distribution of covariates conditioned on the labels remains fixed. For instance, during a pandemic, a disease may spread to a much larger fraction of the population, but the manifestations of the disease may remain the same. As another example, real-world data may have imbalanced classes, unlike the balanced classes typical of curated training datasets. We consider the unsupervised domain adaptation setting (Ben-David et al., 2006), where we are given labeled examples from a *source domain*, but only unlabeled examples from the *target domain*, and care about performance in the target domain.

A standard way to adapt conformal inference to handle distribution shift is by using importance weighting to "convert" data from the source distribution into data from the target distribution (Tibshirani et al., 2019). In the label shift setting, one possible way to express the importance weights is $w^* = \mathbf{C}_P^{-1} q^*$, where $\mathbf{C}_P$ is the confusion matrix and $q^*$ is the distribution of predicted labels (Lipton et al., 2018); see details below. However, the estimation error for the unknown $\mathbf{C}_P$ and $q^*$ breaks the PAC guarantee.

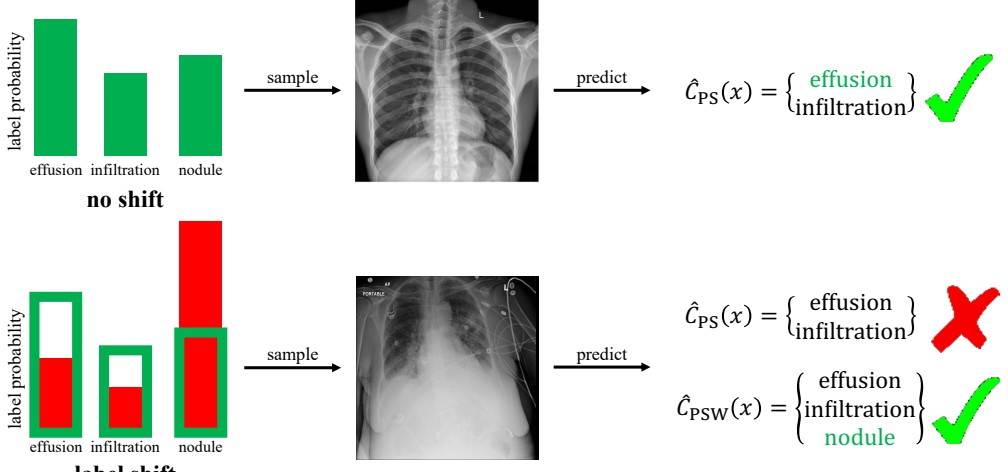

Figure 1: An example of our approach on the ChestX-ray dataset. In the unshifted setting, standard PAC prediction sets guarantee high-probability coverage, but this guarantee fails under label shift. Our approach addresses this challenge and continues to work in the shifted environment.

Instead, we construct confidence intervals around $\mathbf{C}_P$ and $q^*$, and then devise a novel algorithm to propagate these intervals through the Gaussian elimination algorithm used to compute $w^*$. Finally, we leverage an existing strategy for constructing PAC prediction sets when given confidence intervals for the importance weights (Park et al., 2021).

We empirically evaluate our approach on four datasets across three application domains: CIFAR-10 (Krizhevsky et al., 2009) in the computer vision domain, the CDC Heart Dataset (Centers for Disease Control and Prevention (CDC), 1984) and ChestX-ray (National Institutes of Health and others, 2022) in the medical domain, and AGNews (Zhang et al., 2015) in the language domain.

**Contributions.** We propose a novel algorithm for constructing PAC prediction sets in the presence of label shift, which computes provably valid intervals around the true importance weights. Our algorithm is based on a technique for propagating confidence intervals through the updates of Gaussian elimination, which to our knowledge is a novel approach to uncertainty propagation in a prediction set construction setting. Finally, we empirically demonstrate that our approach satisfies the PAC guarantee while constructing smaller prediction sets than several baselines.

**Example.** Figure 1 illustrates a use case of our technique on the ChestX-ray dataset. In medical settings, prediction sets (denoted PS) provide a rigorous way to quantify uncertainty for making downstream decisions. In particular, they can guarantee that the prediction set contains the true label (in this case, a diagnosis) with high probability. However, label shift happens commonly in medical settings, for instance, many illnesses have varying rates of incidence over time even when the patient population remains the same. Unfortunately, label shift breaks the PAC coverage guarantee. Our approach (denoted PSW) corrects for the label shift via importance weighting; it does so in a provably correct way. The resulting prediction sets satisfy the PAC guarantee.

**Related work.** There has been recent interest in conformal inference under distribution shift, much of it focusing on covariate shift (Tibshirani et al., 2019; Lei & Candès, 2021; Qiu et al., 2022). (Podkopaev & Ramdas, 2021) develop methods for marginal coverage under label shift, whereas we are interested in training-set conditional—or PAC—guarantees. Furthermore, they formally assume that the true importance weights are known exactly, which is rarely the case. In the label shift setting, the importance weights can be estimated (Lipton et al., 2018), but as we show in our experiments, uncertainty in these estimates must be handled for the PAC guarantee to hold.

We leverage the method of (Park et al., 2021) to handle estimation error in the importance weights. That work studies covariate shift, and uses a heuristic to obtain intervals around the importance weights. For the label shift setting, we can in fact obtain stronger guarantees: we modify Gaussian

elimination to propagate uncertainty through the computation of the weights $w^* = \mathbf{C}_P^{-1} q^*$. We give a more comprehensive discussion of related work in Appendix A.

## 2 PROBLEM FORMULATION

### 2.1 BACKGROUND ON LABEL SHIFT

Consider the goal of training a classifier $g : \mathcal{X} \to \mathcal{Y}$, where $\mathcal{X} \subseteq \mathbb{R}^d$ is the covariate space, and $\mathcal{Y} = [K] = \{1, ..., K\}$ is the set of labels. We consider the setting where we train on one distribution $P$ over $\mathcal{X} \times \mathcal{Y}$—called the *source*—with a probability density function (PDF) $p : (x, y) \mapsto p(x, y)$, and evaluate on a potentially different test distribution $Q$—called the *target*—with PDF $q : (x, y) \mapsto q(x, y)$. We focus on the unsupervised domain adaptation setting (Ben-David et al., 2007), where we are given an i.i.d. sample $S_m \sim P^m$ of $m$ labeled datapoints, and an i.i.d. sample of $n$ unlabeled datapoints $T_X^n \sim Q_X^n$. The label shift setting (Lipton et al., 2018) assumes that only the label distribution $Q_Y$ may be change from $P_Y$, and the conditional covariate distributions remain the same:

**Assumption 2.1.** (Label shift) We have $p(x \mid y) = q(x \mid y)$ for all $x \in \mathcal{X}, y \in \mathcal{Y}$.

We denote $p(y) = P_Y(Y = y)$ for all $y \in \mathcal{Y}$ and analogously for $Q$. (Lipton et al., 2018) consider two additional mild assumptions:

**Assumption 2.2.** For all $y \in \mathcal{Y}$ such that $q(y) > 0$, we have $p(y) > 0$.

Next, given the trained classifier $g : \mathcal{X} \to \mathcal{Y}$ let $\mathbf{C}_P \in \mathbb{R}^{K \times K}$ denote its expected confusion matrix—i.e., $c_{ij} := (\mathbf{C}_P)_{ij} = \mathbb{P}_{(X,Y) \sim P}(g(X) = i, Y = j)$.

**Assumption 2.3.** *The confusion matrix $\mathbf{C}_P$ is invertible.*

This last assumption requires that the per-class expected predictor outputs be linearly independent; for instance, it is satisfied when $g$ is reasonably accurate across all labels. In addition, one may test whether this assumption holds (Lipton et al., 2018).

Denoting the importance weights $w^* := (q(y)/p(y))_{y \in \mathcal{Y}} \in \mathbb{R}^K$, and $\hat{y} := g(x)$, we will write $p(\hat{y}|y) = \mathbb{P}_{(X,Y) \sim P}[g(X) = \hat{y}|Y = y]$, and define $p(\hat{y}, y)$, $p_{\hat{y}}$ as well as the corresponding expressions for $q$ analogously. Since $\hat{y}$ depends only on $x$, we have $q(\hat{y} \mid y) = p(\hat{y} \mid y)$. Thus, see e.g., Lipton et al. (2018),

$$q_{\hat{y}} = \sum_{y \in \mathcal{Y}} q(\hat{y} \mid y) q(y) = \sum_{y \in \mathcal{Y}} p(\hat{y} \mid y) q(y) = \sum_{y \in \mathcal{Y}} p(\hat{y}, y) \frac{q(y)}{p(y)},$$

or in a matrix form, $q^* = \mathbf{C}_P w^*$, where $q^* := (q_{\hat{y}})_{\hat{y} \in \mathcal{Y}} \in \mathbb{R}^K$. As we assume $\mathbf{C}_P$ is invertible,

$$w^* = \mathbf{C}_P^{-1} q^*. \tag{1}$$

Our algorithm uses this equation to approximate $w^*$, and then use its approximation to construct PAC prediction sets that remain valid under label shift.

### 2.2 PAC PREDICTION SETS UNDER LABEL SHIFT

We are interested in constructing a *prediction set* $C : \mathcal{X} \to 2^{\mathcal{Y}}$, which outputs a set of labels $C(x) \subseteq \mathcal{Y}$ for each given input $x \in \mathcal{X}$ rather than a single label. The benefit of outputting a set of labels is that we can obtain correctness guarantees such as:

$$\mathbb{P}_{(X,Y) \sim P}[Y \in C(X)] \geq 1 - \varepsilon, \tag{2}$$

where $\varepsilon \in (0, 1)$ is a user-provided confidence level. Then, downstream decisions can be made in a way that accounts for all labels $y \in C(x)$ rather than for a single label. Thus, prediction sets quantify uncertainty. Intuitively, equation 2 can be achieved if we output $C(x) = \mathcal{Y}$ for all $x \in \mathcal{X}$, but this is not informative. Instead, the typical goal is to output prediction sets that are as small as possible.

The typical strategy for constructing prediction sets is to leverage an fixed existing model. In particular, we assume given a *scoring function* $f : \mathcal{X} \times \mathcal{Y} \to \mathbb{R}$; most deep learning algorithms provide such scores in the form of predicted probabilities, with the corresponding classifier being $g(x) =$

$\arg\max_{y \in \mathcal{Y}} f(x, y)$. The scores do not need to be reliable in any way; if they are unreliable, the PAC prediction set algorithm will output larger sets. Then, we consider prediction sets parameterized by a real-valued threshold $\tau \in \mathbb{R}$:

$$C_\tau(x) = \{y \in \mathcal{Y} \mid f(x, y) \geq \tau\}.$$

In other words, we include all labels with score at least $\tau$. First, we focus on correctness for $P$, in which case we only need $S_m$, usually referred to as the calibration set. Then, a prediction set algorithm constructs a threshold $\hat{\tau}(S_m) \in \mathbb{R}$ and returns $C_{\hat{\tau}(S_m)}$.

Finally, we want $\hat{\tau}$ to satisfy (2); one caveat is that it may fail to do so due to randomness in $S_m$. Thus, we allow an additional probability $\delta \in \mathbb{R}$ of failure, resulting in the following desired guarantee:

$$\mathbb{P}_{S_m \sim P^m}[\mathbb{P}_{(X,Y) \sim P}[Y \in C_{\hat{\tau}(S_m)}(X)] \geq 1 - \varepsilon] \geq 1 - \delta. \tag{3}$$

Vovk (2012); Park et al. (2019) propose an algorithm $\hat{\tau}$ that satisfies (3), see Appendix B.

Finally, we are interested in constructing PAC prediction sets in the label shift setting, using both the labeled calibration dataset $S_m \sim P^m$ from the source domain, and the unlabeled calibration dataset $T_n^X \sim Q^n$ from the target distribution. Our goal is to construct $\hat{\tau}(S_m, T_n^X)$ based on both $S_m$ and $T_n^X$, which satisfies the coverage guarantee over $Q$ instead of $P$:

$$\mathbb{P}_{S_m \sim P^m, T_n^X \sim Q_X^n}\left[\mathbb{P}_{(X,Y) \sim Q}[Y \in C_{\hat{\tau}(S_m, T_n^X)}(X)] \geq 1 - \varepsilon\right] \geq 1 - \delta. \tag{4}$$

Importantly, the inner probability is over the shifted distribution $Q$ instead of $P$.

## 3 ALGORITHM

To construct prediction sets valid under label shift, we first notice that it is enough to find element-wise confidence intervals for the importance weights $w^*$. Suppose that we can construct $W = \prod_{k \in \mathcal{Y}} [\underline{w}_k, \overline{w}_k] \subseteq \mathbb{R}^K$ such that $w^* \in W$. Then, when adapted to our setting, the results of Park et al. (2021)—originally for the covariate shift problem—provide an algorithm that returns a threshold $\hat{\tau}(S_m, V, W, b)$, where $V \sim \text{Uniform}([0, 1])^K$ is a vector of random variables, such that

$$\mathbb{P}_{S_m \sim P^m, V \sim U^K}\left[\mathbb{P}_{(X,Y) \sim Q}[Y \in C_{\hat{\tau}(S_m, V, W, b)}] \geq 1 - \varepsilon\right] \geq 1 - \delta. \tag{5}$$

This is similar to equation 4 but it accounts for the randomness used by our algorithm—via $V$—in the outer probability. We give the details in Appendix C.

The key challenge is to construct $W = \prod_{k \in \mathcal{Y}} [\underline{w}_k, \overline{w}_k]$ such that $w^* \in W$ with high probability. The approach from Park et al. (2021) for the covariate shift problem relies on training a source-target discriminator, which is not possible in our case since we do not have class labels from the target domain. Furthermore, Park et al. (2021)'s approach is does not provide conditions under which one can provide a valid confidence interval for the importance weights in their setting.

Our algorithm uses a novel approach, where we propagate intervals through the computation of importance weights. The weights $w^*$ are determined by the system of linear equations $\mathbf{C}_P w^* = q^*$. Since $\mathbf{C}_P$ and $q^*$ are unknown, we start by constructing *element-wise* confidence intervals

$$\underline{\mathbf{C}}_P \leq \mathbf{C}_P \leq \overline{\mathbf{C}}_P \qquad \text{and} \qquad \underline{q}^* \leq q^* \leq \overline{q}^*, \tag{6}$$

with probability at least $1 - \delta$ over our calibration datasets $S_m$ and $T_n^X$. We then propagate these confidence intervals through each step of Gaussian elimination, such that at the end of the algorithm, we obtain confidence intervals for its output—i.e.,

$$\underline{w}^* \leq w^* \leq \overline{w}^* \qquad \text{with probability at least } 1 - \delta. \tag{7}$$

Finally, we can use (7) with the algorithm from (Park et al., 2021) to construct PAC prediction sets under label shift. We describe our approach below.

### 3.1 ELEMENTWISE CONFIDENCE INTERVALS FOR $\mathbf{C}_P$ AND $q^*$

Recall that $\mathbf{C}_P = (c_{ij})_{ij \in \mathcal{Y}}$ and $q^* = (q_k)_{k \in \mathcal{Y}}$. Note that $c_{ij} = \mathbb{P}[g(X) = i, Y = j]$ is the mean of the Bernoulli random variable $\mathbb{1}(g(X) = i, Y = j)$ over the randomness in $(X, Y) \sim P$.

Similarly, $q_k$ is the mean of $\mathbb{1}(g(X) = k)$ over the randomness in $X \sim Q_X$. Thus, we can use the Clopper-Pearson (CP) intervals (Clopper & Pearson, 1934) for a Binomial success parameter to construct intervals around $c_{ij}$ and $q_k$. Given a confidence level $\delta \in (0, 1)$ and the sample mean $\hat{c}_{ij} = \frac{1}{m} \sum_{(x,y) \in S_m} \mathbb{1}(g(x) = i, y = j)$—distributed as a scaled Binomial random variable—this is an interval $\mathrm{CP}(\hat{c}_{ij}, m, \delta) = [\underline{c}_{ij}, \overline{c}_{ij}]$ such that

$$\mathbb{P}_{S_m \sim P^m}[c_{ij} \in \mathrm{CP}(\hat{c}_{ij}, m, \delta)] \geq 1 - \delta.$$

Similarly, for $q_k$, we can construct CP intervals based on $\hat{q}_k = \frac{1}{n} \sum_{x \in T_n^X} \mathbb{1}(g(x) = k)$. Together, for confidence levels $\delta_{ij}$ and $\delta_k$ chosen later, we obtain for all $i, j, k \in [K]$,

$$\mathbb{P}_{S_m \sim P^m} \left[ \underline{c}_{ij} \leq c_{ij} \leq \overline{c}_{ij} \right] \geq 1 - \delta_{ij}, \qquad \mathbb{P}_{T_n^X \sim Q_X^n} \left[ \underline{q}_k \leq q_k \leq \overline{q}_k \right] \geq 1 - \delta_k. \tag{8}$$

Then, the following result holds by a union bound: Given any $\delta_{ij}, \delta_k \in (0, \infty)$, for all $i, j, k \in [K]$, letting $[\underline{c}_{ij}, \overline{c}_{ij}] = \mathrm{CP}(\hat{c}_{ij}, m, \delta_{ij})$ and $[\underline{q}_k, \overline{q}_k] = \mathrm{CP}(\hat{q}_k, n, \delta_k)$, and letting $\delta = \sum_{i,j \in [K]} \delta_{ij} + \sum_{k \in [K]} \delta_k$, we have

$$\mathbb{P}_{S_m \sim P^m, T_n^X \sim Q_X^n} \left[ \bigwedge_{i,j \in [K]} \underline{c}_{ij} \leq c_{ij} \leq \overline{c}_{ij}, \bigwedge_{k \in [K]} \underline{q}_k \leq q_k \leq \overline{q}_k \right] \geq 1 - \delta. \tag{9}$$

### 3.2 GAUSSIAN ELIMINATION WITH INTERVALS

We also need to set up notation for Gaussian elimination, which requires us to briefly recall the algorithm. To solve $\mathbf{C}_P w^* = q^*$, Gaussian elimination (see e.g., Golub & Van Loan, 2013) proceeds in two phases. Starting with $c^0 = \mathbf{C}_P$ and $q^0 = q^*$, on iteration $t \geq 1$, Gaussian elimination uses row $k = t$ to eliminate the $k$th column of rows $i \in \{k + 1, ..., K\}$ (we introduce a separate variable $k$ for clarity). In particular, if $c_{kk}^t \neq 0$, we denote

$$c_{ij}^{t+1} = \begin{cases} c_{ij}^t - \dfrac{c_{ik}^t c_{kj}^t}{c_{kk}^t} & \text{if } i > k, \\ c_{ij}^t & \text{otherwise;} \end{cases} \qquad q_i^{t+1} = \begin{cases} q_i^t - \dfrac{c_{ik}^t q_k^t}{c_{kk}^t} & \text{if } i > k \\ q_i^t & \text{otherwise,} \end{cases} \qquad \forall i, j \in [K].$$

If $c_{kk}^t = 0$, but there is an element $j > k$ in the $k$th column such that $c_{jk}^t \neq 0$, the $k$th and the $j$th rows are swapped and the above steps are executed. If no such element exists, the algorithm proceeds to the next step. At the end of the first phase, the matrix $c^{K-1}$ has all elements below the diagonal equal to zero—i.e., $c_{ij}^{K-1} = 0$ if $j < i$. In the second phase, the Gaussian elimination algorithm solves for $w^*$ backwards from $i = K$ to $i = 1$, introducing the following notation. For each $i$, if $c_{ii}^{K-1} \neq 0$, we denote[1] $w_i^* = (q_i - s_i)/c_{ii}^{K-1}$, where $s_i = \sum_{j=i+1}^{K} c_{ij}^{K-1} w_j^*$.

In our setting, we do not know $c^0$ and $q^0$; instead, we assume given entrywise confidence intervals as in equation 6, which amount to $\underline{c}^0 \leq c^0 \leq \overline{c}^0$ and $\underline{q}^0 \leq q^0 \leq \overline{q}^0$. We now work on the event $\Omega$ that these bounds hold, and prove that our algorithm works on this event; later, we combine this result with Equation 9 to obtain a high-probability guarantee. Then, our goal is to compute $\underline{c}^t, \overline{c}^t, \underline{q}^t, \overline{q}^t$ such that for all iterations $t \in \{0, 1, ..., K - 1\}$, we have elementwise confidence intervals specified by $\underline{c}^t, \overline{c}^t, \underline{q}^t$ and $\overline{c}^t$ for the outputs $c^t, q^t$ of the Gaussian elimination algorithm:

$$\underline{c}^t \leq c^t \leq \overline{c}^t \qquad \text{and} \qquad \underline{q}^t \leq q^t \leq \overline{q}^t. \tag{10}$$

The base case $t = 0$ holds by the assumption. Next, to propagate the uncertainty through the Gaussian elimination updates for each iteration $t \in [K - 1]$, our algorithm sets

$$\underline{c}_{ij}^{t+1} = \begin{cases} 0 & \text{if } i > k, \ j \leq k, \\ \underline{c}_{ij}^t - \dfrac{\overline{c}_{ik}^t \overline{c}_{kj}^t}{\underline{c}_{kk}^t} & \text{if } i, j > k, \\ \underline{c}_{ij}^t & \text{otherwise} \end{cases} \qquad \forall i, j \in [K] \tag{11}$$

---

[1]The algorithm requires further discussion if $c_{ii}^{K-1} = 0$ (Golub & Van Loan, 2013); this does not commonly happen in our motivating application so we will not consider this case. See Appendix D for details.

for the lower bound, and computes

$$\bar{c}_{ij}^{t+1} = \begin{cases} 0 & \text{if } i > k,\ j \le k, \\ \bar{c}_{ij}^t - \dfrac{\underline{c}_{ik}^t \underline{c}_{kj}^t}{\bar{c}_{kk}^t} & \text{if } i, j > k, \\ \bar{c}_{ij}^t & \text{otherwise} \end{cases} \qquad \forall i, j \in [K] \tag{12}$$

for the upper bound. The first case handles the fact that Gaussian elimination is guaranteed to zero out entries below the diagonal, and thus these entries have no uncertainty remaining. The second rule constructs confidence intervals based on the previous intervals and the algebraic update formulas used in Gaussian elimination for the entries for which $i, j > k$. For instance, the above confidence intervals use that on the event $\Omega$, and by induction on $t$, if $\underline{c}_{ij}^t \ge 0$ and $\underline{c}_{ii}^t > 0$ for all $i, j \in [K]$ and for all $t$, the Gaussian elimination update $c_{ij}^{t+1} = c_{ij}^t - c_{it}^t c_{tj}^t / c_{tt}^t$ can be upper bounded as

$$c_{ij}^{t+1} = c_{ij}^t - \frac{c_{it}^t c_{tj}^t}{c_{tt}^t} \le \bar{c}_{ij}^t - \frac{\underline{c}_{it}^t \underline{c}_{tj}^t}{\bar{c}_{tt}^t} = \bar{c}_{ij}^{t+1}, \tag{13}$$

The assumptions that $\underline{c}_{ij}^t \ge 0$ and $\underline{c}_{ii}^t > 0$ for all $i, j \in [K]$ and for all $t$ may appear a little stringent, but the former can be removed at the cost of slightly larger intervals propagated to the next step, see Section D. The latter condition is satisfied by any classifier that obtains sufficient accuracy on all labels. We further discuss these conditions in Section D. The third rule in equation 11 and equation 12 handles the remaining entries, which do not change; and thus the confidence intervals from the previous step can be used. The rules for $q$ are similar, and have a similar justification:

$$\underline{q}_i^{t+1} = \begin{cases} \underline{q}_i^t - \dfrac{\bar{c}_{ik}^t \bar{q}_i^t}{\underline{c}_{kk}^t} & \text{if } i > k, \\ \underline{q}_i^t & \text{otherwise}; \end{cases} \qquad \bar{q}_i^{t+1} = \begin{cases} \bar{q}_i^t - \dfrac{\underline{c}_{ik}^t \underline{q}_i^t}{\bar{c}_{kk}^t} & \text{if } i > k, \\ \bar{q}_i^t & \text{otherwise}. \end{cases} \qquad \forall i \in [K]. \tag{14}$$

For these rules, our algorithm assumes $\underline{q}_i^t \ge 0$ for all $i \in [K]$ and all $t$, and raises an error if this fails. As with the first condition above, this one can be straightforwardly relaxed; see Appendix D.

In the second phase, we compute $w_i^*$ starting from $i = K$ and iterating to $i = 1$. On iteration $i$, we assume we have the confidence intervals $\underline{w}_j^* \le w_j^* \le \overline{w}_j^*$ for $j > i$. Then, we compute confidence intervals for the sum $s_i$, with a similar justification based on the Gaussian elimination updates:

$$\underline{s}_i = \sum_{j=i+1}^n \underline{c}_{ij}^{K-1} \underline{w}_j^* \qquad \text{and} \qquad \bar{s}_i = \sum_{j=i+1}^n \bar{c}_{ij}^{K-1} \overline{w}_j^*, \tag{15}$$

and show that they satisfy $\underline{s}_i \le s_i \le \bar{s}_i$ on the event $\Omega$. Finally, we compute confidence intervals for $w_i^*$, assuming $\underline{c}_{ii}^{K-1} > 0$:

$$\underline{w}_i^* = \frac{\underline{q}_i - \bar{s}_i}{\bar{c}_{ii}^{K-1}} \qquad \text{and} \qquad \overline{w}_i^* = \frac{\bar{q}_i - \underline{s}_i}{\underline{c}_{ii}^{K-1}}, \tag{16}$$

for which we can show that they satisfy $\underline{w}_i^* \le w_i^* \le \overline{w}_i^*$ based on the Gaussian elimination updates. Letting $W = \{ w \mid \underline{w}^* \le w \le \overline{w}^* \}$, we have the following (see Appendix E for a proof).

**Lemma 3.1** (Elementwise Confidence Interval for Importance Weights). *If (6) holds, and for all $i, j, t \in [K]$, $\underline{c}_{ij}^t \ge 0$, $\underline{c}_{ii}^t > 0$, and $\underline{q}_i^t \ge 0$, then $w^* = \mathbf{C}_P^{-1} q^* \in W$.*

We mention here that the idea of algorithmic uncertainty propagation may be of independent interest. In future work, it may further be developed to other methods for solving linear systems (e.g., the LU decomposition, Golub & Van Loan (2013)), and other linear algebraic and numerical computations.

## 3.3 Overall Algorithm

Algorithm 1 summarizes our approach. As can be seen, the coverage levels for the individual Clopper-Pearson intervals are chosen to satisfy the overall $1 - \delta$ coverage guarantee. In particular, the PAC guarantee equation 4 follows from equation 5, equation 9, Lemma 3.1, and a union bound; we provide a more detailed explanation in Appendix F.

**Theorem 3.2** (PAC Prediction Sets under Label Shift). *For any given $\varepsilon, \delta \in (0, 1)$, under Assumptions 2.1, 2.2 and 2.3, if $\forall i, j, t \in [K]$, we have $\underline{c}_{ij}^t \ge 0$, $\underline{c}_{ii}^t > 0$, and $\underline{q}_i^t \ge 0$, then Algorithm 1 satisfies*

$$\mathbb{P}_{S_m \sim P^m, T_n^X \sim Q^n, V \sim U^m} \left[ \mathbb{P}_{(X,Y) \sim Q}[Y \in C_{\hat{\tau}(S_m, V, W, b)}(X)] \ge 1 - \varepsilon \right] \ge 1 - \delta.$$

As discussed in Appendix D, we can remove the requirement that $\underline{c}_{ij}^t \ge 0$ and $\underline{q}_i^t \ge 0$ for $i \ne j$.

---

**Algorithm 1** PS-W: PAC prediction sets in the label shift setting.

---

1: **procedure** LABELSHIFTPREDICTIONSET($S_m, T_n^X, f, \varepsilon, \delta$)
2: $\quad \underline{c}, \overline{c}, \underline{q}, \overline{q} \leftarrow$ CPINTERVAL($S_m, T_n^X, x \mapsto \arg\max_{y \in \mathcal{Y}} f(x,y), \frac{K(K+1)}{(K(K+1)+1)}\delta$)
3: $\quad W \leftarrow$ INTERVALGAUSSIANELIM($\underline{c}, \overline{c}, \underline{q}, \overline{q}$)
4: $\quad$ **if** $W = \varnothing$ **then return** $\varnothing$
5: $\quad$ **return** IWPREDICTIONSET($S_m, f, W, \varepsilon, \delta/[K(K+1)+1]$)
6: **procedure** CPINTERVAL($S_m, T_n^X, g, \delta$)
7: $\quad$ **for** $i, j \in [K]$ **do**
8: $\qquad$ Compute $[\underline{c}_{ij}, \overline{c}_{ij}] = \mathrm{CP}\left(m^{-1}\sum_{(x,y)\in S_m} \mathbb{1}(g(x)=i, y=j), m, \delta/(K(K+1))\right)$
9: $\quad$ **for** $k \in [K]$ **do**
10: $\qquad$ Compute $[\underline{q}_k, \overline{q}_k] = \mathrm{CP}\left(n^{-1}\sum_{x\in T_n^X} \mathbb{1}(g(x)=k), n, \delta/(K(K+1))\right)$
11: $\quad$ **return** $\underline{c}, \overline{c}, \underline{q}, \overline{q}$
12: **procedure** INTERVALGAUSSIANELIM($\underline{c}^0, \overline{c}^0, \underline{q}^0, \overline{q}^0$)
13: $\quad$ **for** $t \in [1, ..., K-1]$ **do**
14: $\qquad$ **for** $i, j \in [K]$ **do**
15: $\qquad\quad$ Compute $\underline{c}_{ij}^t, \overline{c}_{ij}^t$ using (11) & (12), and $\underline{q}_i^t, \overline{q}_i^t$ using (14)
16: $\qquad\quad$ **if** $\underline{c}_{ij}^t < 0$ for some $i \neq j$ or $\underline{c}_{ii}^t \leq 0$ for some $i$ **or** $\underline{q}_i^t \leq 0$ for some $i$, **then return** $\varnothing$
17: $\qquad$ **for** $i \in [K, ..., 1]$ **do**
18: $\qquad\quad$ Compute $\underline{s}_i^t, \overline{s}_i^t$ using (15), and $\underline{w}_i, \overline{w}_i$ using (16)
19: $\quad$ **return** $W = \prod_{i=1}^k [\underline{w}_i, \overline{w}_i]$
20: **procedure** IWPREDICTIONSET($S_m, f, W = \prod_{k=1}^K [\underline{w}_k, \overline{w}_k], \varepsilon, \delta$)
21: $\quad V \sim$ Uniform($[0,1]$)$^m$
22: $\quad$ **return** $\hat{\tau}(S_m, V, W, \max_{k\in[K]} \overline{w}_k, \varepsilon, \delta)$ as in (18)

---

# 4 EXPERIMENTS

## 4.1 EXPERIMENTAL SETUP

**Predictive models.** We analyze four datasets: the CDC Heart dataset, CIFAR-10, Chest X-ray, and the AG News dataset; details are provided in Section 4.2. We use a two-layer MLP for the CDC Heart data with an SGD optimizer having a learning rate of 0.03 and a momentum of 0.9, using a batch size of 64 for 30 epochs. For CIFAR-10, we finetune a pretrained ResNet50 He et al. (2016), with a learning rate of 0.01 for 56 epochs. For the ChestX-ray14 dataset, we use a pre-trained CheXNet (Rajpurkar et al., 2017) with a DenseNet121 (Huang et al., 2017) backbone with a learning rate of 0.0003 for two epochs. For AGNews, we fine-tune a pre-trained Electra sequence classifier for one epoch with an AdamW optimizer using a learning rate of 0.00001.

**Hyperparameter choices.** There are two user-specified hyperparameters that control the guarantees, namely $\delta$ and $\varepsilon$. In our experiments, we choose $\delta = 5 \times 10^{-4}$ to ensure that, over 100 independent datasets $S_m$, there is a 95% probability that the error rate is not exceeded. Specifically, this ensures that $\mathbb{P}_{(X,Y)\sim P}[Y \in C_{\hat{\tau}(S_m)}(X)] \geq 1 - \varepsilon$ holds for all 100 trials, with probability approximately $1 - 0.95^{1/100} \approx 5 \times 10^{-4}$. We select $\varepsilon$ for each dataset in a way that the resulting average prediction set size is greater than one. This ensures that the coverage guarantee is non-trivial, as a single-valued classifier should not achieve the desired coverage rate (as $\delta$ is small).

**Dataset construction.** We follow the label shift simulation strategies from previous work (Lipton et al., 2018). First, we split the full dataset into training data, and "base" source and target datasets. We use the training dataset to fit a score function. Given label distributions $P_Y$ and $Q_Y$, we generate the source dataset $S_m$, target dataset $T_n^X$, and a labeled, size $o$ target test dataset (sampled from $Q$) by sampling with replacement from the corresponding base datasets. We consider two choices of $P_Y$ and $Q_Y$: (i) a **tweak-one** shift, where change the probability of one label, and keep the relative proportions of the other labels equally likely, and (ii) a **general** shift, where we shift each probability as described later.

**Baselines.** We compare our approach (**PS-W**) with several baselines (see Appendix G):

- **PS:** PAC prediction sets that do not account for label shift (Vovk, 2012; Park et al., 2019). This does not come with PAC guarantees under label shift.
- **WCP:** Weighted conformal prediction under label shift, which targets marginal coverage (Pod-kopaev & Ramdas, 2021). This does not come with PAC guarantees under label shift either.
- **PS-R:** PAC prediction sets that account for label shift but ignore uncertainty in the importance weights; again without guarantees.
- **PS-C:** This addresses label shift via a conservative upper bound on the empirical loss (see Appendix G for details). This is the only baseline with a PAC guarantee under label shift.

We compare to other baselines in Appendix I.5, and to an oracle with the true weights in Appendix H. Results for other hyperparameters are in Appendix I.

**Metrics.** We measure performance via the prediction set error, i.e., the fraction of $(x, y) \sim Q$ such that $y \notin C_\tau(x)$; and the average prediction set size, i.e., the mean of $|C_\tau(x)|$, evaluated on the held-out test set. We report the results over 100 independent repetitions, randomizing both dataset generation and our algorithm.

## 4.2 RESULTS & DISCUSSION

**CDC Heart.** We use the CDC Heart dataset, a binary classification problem (Centers for Disease Control and Prevention (CDC), 1984). The goal is to predict the risk of heart attack given features such as level of exercise or weight. We use $\varepsilon = 0.1$ and $\delta = 5 \times 10^{-4}$. We consider both large and small shifts. For the large shift, the label distributions—denoted (pos%, neg%)—are $(94\%, 6\%)$ for

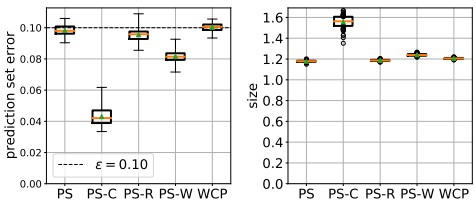
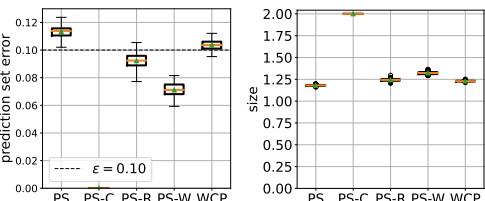

(a) Prediction set error and size under small shifts on the CDC Heart dataset. Parameters are $\varepsilon = 0.1$, $\delta = 5 \times 10^{-4}$, $m = 42000$, $n = 42000$, and $o = 9750$.

(b) Prediction set error and size under large shifts on the CDC Heart dataset, Parameters are $\varepsilon = 0.1$, $\delta = 5 \times 10^{-4}$, $m = 42000$, $n = 42000$, and $o = 9750$.

Figure 2: Prediction set results on the CDC Heart dataset.

source, and $(63.6\%, 36.4\%)$ for target; results are in Figure 2b. We also consider a small shift with label distributions $(94\%, 6\%)$ for source, and $(91.3\%, 8.7\%)$ for target; results are in Figure 2a. As can be seen, our PS-W algorithm satisfies the PAC guarantee while achieving smaller prediction set size than PS-C, the only baseline to satisfy the PAC guarantee. The PS and PS-R algorithms violate the PAC guarantee.

**CIFAR-10.** Next, we consider CIFAR-10 (Krizhevsky et al., 2009), which has 10 labels. We use $\varepsilon = 0.1$ and $\delta = 5 \times 10^{-4}$. We consider a large and a small tweak-one shift. For the large shift, the label probability is 10% for all labels in the source, 40.0% for the tweaked label, and 6.7% for other labels in the target; results are in Figure 3a. For small shifts, we use 10% for all labels for the source, 11.76% for the tweaked label, and 9.8% for other labels for the target; results are in Figure 3b. Under large shifts, our PS-W algorithm satisfies the PAC guarantee while outperforming PS-C by a large margin. When the shift is very small, PS-W still satisfies the PAC guarantee, but generates more conservative prediction sets similar in size to those of PS-C (e.g., Figure 3b) given the limited data. Results for a non-uniform source distribution and general shifts are shown in Appendix I.8.

**AGNews.** AG News is a subset of AG's corpus of news articles (Zhang et al., 2015). It is a text classification dataset with four labels: World, Sports, Business, and Sci/Tech. It contains 31,900 unique examples for each class. We use $\varepsilon = 0.05$ and $\delta = 5 \times 10^{-4}$. We use tweak-one label shifts. We consider a large shift and a medium-sized calibration dataset, with label distributions equalling $(30.8\%, 30.8\%, 7.7\%, 30.8\%)$ for the source, and $(12.5\%, 12.5\%, 62.5\%, 12.5\%)$ for the target; results are in Figure 4a. As before, our PS-W approach satisfies the PAC guarantee while achieving smaller set sizes than PS-C.

**ChestX-ray.** ChestX-ray14 (Wang et al., 2017) is a medical imaging dataset containing about 112K frontal-view X-ray images of 30K unique patients with fourteen disease labels. This dataset contains

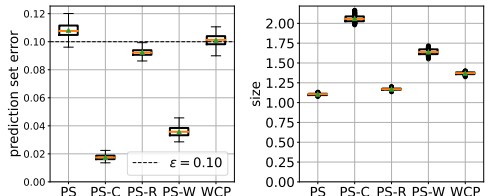 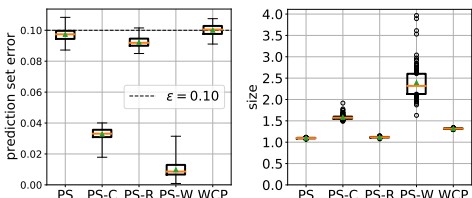

(a) Prediction set error and size with larger shift on the CIFAR-10. Parameters are $\varepsilon = 0.1$, $\delta = 5 \times 10^{-4}$, $m = 27000$, $n = 19997$, and $t = 19997$.

(b) Prediction set error and size with small shift on the CIFAR-10. Parameters are $\varepsilon = 0.1$, $\delta = 5 \times 10^{-4}$, $m = 27000$, $n = 16500$, and $t = 16500$.

Figure 3: Prediction set results on the CIFAR-10 dataset.

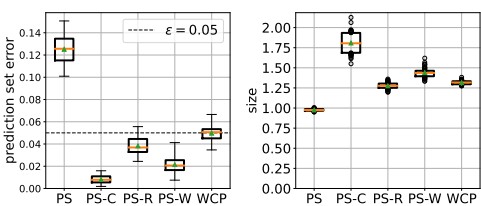 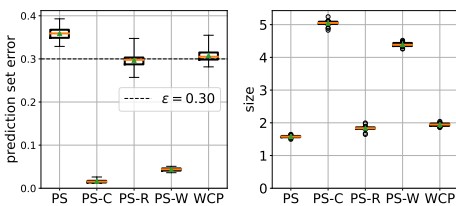

(a) Prediction set error and size on the AGNews Dataset. Parameters are $\varepsilon = 0.05$, $\delta = 5 \times 10^{-4}$, $m = 26000$, $n = 12800$, and $t = 12800$.

(b) Prediction set error and size on the ChestX-ray dataset. Parameters are $\varepsilon = 0.3$, $\delta = 5 \times 10^{-4}$, $m = 67200$, $n = 35200$, and $o = 3520$.

Figure 4: Prediction set results on the ChestX-ray dataset.

instances with multiple labels, which we omit. We also omit classes with few positively labeled datapoints, leaving six classes: Atelectasis, Effusion, Infiltration, Mass, Nodule, Pneumothorax. We consider a general label shift. We consider a large tweak-one shift, with label distributions of $(19.1\%, \ldots, 19.1\%, 4.5\%, 19.1\%)$ for the source, and $(11.1\%, \ldots, 11.1\%, 44.5\%, 11.1\%)$ for the target. Results for $\varepsilon = 0.3$ are in Figure 4b. As before, our PS-W approach satisfies the PAC guarantee while outperforming PS-C. The PS-R and WCP methods violate the constraint.

**Discussion.** In all our experiments, our approach satisfies the PAC guarantee; furthermore, it produces smaller prediction set sizes than PS-C—the only baseline to consistently satisfy the PAC guarantee— except when the label shift is small and the calibration dataset is limited. In contrast, the PS baseline does not account for label shift, and the PS-R baseline does not account for uncertainty in the importance weights, so they do not satisfy the PAC guarantee. The WCP baseline is designed to target a different guarantee, and it does not satisfy the PAC guarantee. Thus, these results demonstrate the efficacy of our approach.

**Limitations.** Our approach is focused on problem settings where the label shift is not too small and sufficient calibration data is available; and may produce conservative prediction sets otherwise. This reflects the intrinsic difficulty of the problem in these settings. Importantly, our PAC coverage guarantees still hold.

## 5 CONCLUSION

We have proposed a PAC prediction set algorithm for the label shift setting, and illustrated its effectiveness in experiments. Directions for future work include improving performance when the calibration dataset or the label shift is small.

**Reproducibility statement.** Our code is available at `https://github.com/averysi224/pac-ps-label-shift` for reproducing our experiments.

## ACKNOWLEDGEMENT

This work was supported in part by ARO W911NF-20-1-0080 ARO W911NF-23-1-0296, NSF 2031895, NSF 2046874, ONR N00014-21-1-2843, the Sloan Foundation, and Institute of Information & communications Technology Planning & Evaluation (IITP) grant funded by the Korea government (MSIT) (No.2019-0-01906, Artificial Intelligence Graduate School Program (POSTECH)). Any opinions, findings and conclusions or recommendations expressed in this material are those of the authors and do not necessarily reflect the views of the Army Research Office (ARO), or the Department of Defense, or the United States Government.

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

## A   ADDITIONAL RELATED WORK

**Conformal Prediction.** Our work falls into the broad areas of distribution-free uncertainty quantification, conformal prediction and tolerance regions (e.g., Guttman, 1970; Vovk et al., 2005; Balasubramanian et al., 2014; Angelopoulos & Bates, 2021; Kaur et al., 2022; Li et al., 2022; Dobriban & Yu, 2023, etc), which aim to construct prediction sets with finite sample coverage guarantees. The prediction sets are often realized by a setting a threshold on the values of a traditional single-label predictor (i.e. *conformity/non-conformity score function*) and predict all labels with scores above this threshold. Our setting is related to inductive conformal prediction (Papadopoulos et al., 2002; Vovk, 2012; Lei et al., 2015), where the dataset is split into a training set for fitting the scoring function and a calibration set for constructing the prediction sets.

**PAC Prediction Sets.** A standard coverage guarantee for conformal prediction methods is marginal coverage over the training, calibration and test data. For inductive conformal prediction, training-data conditional correctness (Vovk, 2012; Park et al., 2019) aims to achieve an $(\varepsilon, \delta)$-coverage guarantee, where coverage over test points exceeds the desired error rate level $\varepsilon$ with probability at most $\delta$. Sets satisfying this guarantee have been termed PAC prediction sets (Park et al., 2019). This guarantee is also equivalent to the classical coverage properties of tolerance regions (Wilks, 1941; Fraser, 1956). PAC coverage may be desired when a slight excess of the desired error rate cab cause a significant safety concern.

|  | Covariate shift | Label shift |
|---|---|---|
| Importance weights, as a function of $x, y$ | $q(x)/p(x)$ | $q(y)/p(y)$ |
| Shared conditional distribution, for all $x, y$ | $p(y \mid x) = q(y \mid x)$ | $p(x \mid y) = q(x \mid y)$ |

Table 1: Comparison of covariate shift and label shift.

**Label shift.** Label shift (Zadrozny, 2004; Huang et al., 2006; Sugiyama et al., 2007; Gretton et al., 2009) supposes that the conditional covariate distribution is fixed but the label distribution may change from a source to a target distribution; see Table 1. In more detail, let $p$ and $q$ denote the probability density functions of the source and target domains with respect to a common dominating measure, respectively. Label shift assumes that for all labels $y$, $q(y)$ may differ from $p(y)$, but for all features $x$ and labels $y$, we have $p(x \mid y) = q(x \mid y)$. Label shift can arise when the representations of the classes changes, for instance in scenarios like medical diagnosis and object recognition (Storkey et al., 2009; Saerens et al., 2002; Lipton et al., 2018).

Early solutions required the estimation of $q$ and $p$, which may scale poorly with the dimension (Chan & Ng, 2005; Storkey et al., 2009; Zhang et al., 2013). More recently, two approaches achieved scalability by assuming an approximate relationship between ground truth labels $y$ and the outputs $\hat{y}$ of a classifier (Lipton et al., 2018; Azizzadenesheli et al., 2019; Saerens et al., 2002): Black Box Shift Estimation (BBSE) (Lipton et al., 2018) and RLLS (Azizzadenesheli et al., 2019) provided consistency results and finite-sample guarantees assuming the confusion matrix is invertible. Subsequent work developed a unified framework that decomposes the error in computing importance weights into miscalibration error and estimation error, with BBSE as a special case (Garg et al., 2020); this approach was extended to open set label shift domain adaptation (Garg et al., 2022). We also mention that label shift is a special case of a more general class of distribution shifts that allow the source and target distributions to be related by sharing sequential conditional distributions (Qiu et al., 2023).

**Conformal methods and distribution shift.** Conformal prediction has been adapted to distribution shifts such as covariate shift (Tibshirani et al., 2019; Lei & Candès, 2021; Qiu et al., 2022). (Podkopaev & Ramdas, 2021) considers label shift when the true importance weights are exactly known. Importance weights are typically estimated with uncertainty. We address this by developing an algorithm to find confidence intervals for the importance weights. More broadly, prediction sets have been studied in the meta-learning setting (Dunn et al., 2022; Park et al., 2022), as well as the setting of robustness to all distribution shifts with bounded $f$-divergence (Cauchois et al., 2020).

**Class-conditional prediction sets.** Although not designed specifically for solving the label shift problem, methods for class-conditional coverage (Sadinle et al., 2019) and adaptive prediction sets (APS) (Romano et al., 2020) can improve robustness to label shifts. However, class-conditional

coverage is a stronger guarantee that leads to prediction sets larger than for our algorithm, while APS does not satisfy a PAC guarantee; we compare empirically to these approaches in Appendix I.5.

## B  BACKGROUND ON PAC PREDICTION SETS

Finding the maximum $\tau$ that satisfies (3) is equivalent to choosing the largest $\tau$ such that the empirical error

$$\bar{L}_{S_m}(C_\tau) := \sum_{(x,y) \in S_m} \mathbb{1}(y \notin C_\tau(x))$$

on the calibration set $S_m$ is bounded Vovk et al. (2005); Park et al. (2019). Let $F(k; m, \varepsilon) = \sum_{i=0}^{k} \binom{m}{k} \varepsilon^i (1 - \varepsilon)^{m-i}$ be the cumulative distribution function (CDF) of the binomial distribution $\text{Binom}(m, \varepsilon)$ with $m$ trials and success probability $\varepsilon$ evaluated at $k$. Then, Park et al. (2019) constructs $C_{\hat{\tau}}$ via

$$\hat{\tau} = \max_{\tau \in T} \tau \quad \text{subj. to} \quad \bar{L}_{S_m}(C_\tau) \leq k(m, \varepsilon, \delta), \tag{17}$$

$$\text{where} \quad k(m, \varepsilon, \delta) = \max_{k \in \mathbb{N} \cup \{0\}} k \quad \text{subj. to} \quad F(k; m, \varepsilon) \leq \delta.$$

Their approach is equivalent to the method from Vovk (2012), but formulated in the language of learning theory. By viewing the prediction set as a binary classifier, the PAC guarantee via this construction can be connected to the Binomial distribution. Indeed, for fixed $C$, $\bar{L}_{S_m}(C)$ has distribution $\text{Bionm}(m, L_P(C))$, since $\mathbb{1}(y \notin C(x))$ has a $\text{Bernoulli}(L_P(C))$ distribution when $(x, y) \sim P$. Thus, $k(m, \varepsilon, \delta)$ defines a bound such that if $L_P(C) \leq \varepsilon$, then $\bar{L}_{S_m}(C) \leq k(m, \varepsilon, \delta)$ with probability at least $1 - \delta$.

## C  BACKGROUND ON PREDICTION SETS UNDER DISTRIBUTION SHIFT

Here we demonstrate how to obtain prediction sets given intervals $\underline{w}^* \leq w^* \leq \overline{w}^*$ around the true importance weights. This approach is based closely on the strategy in (Park et al., 2021) for constructing prediction sets under covariate shift, but adapts it to the label shift setting (indeed, our setting is simpler since there are finitely many importance weights). The key challenge is computing the importance weight intervals, which we describe in detail below.

Given the true importance weights $w^*$, one strategy would be to use rejection sampling (Von Neumann, 1951; Shapiro, 2003; Rubinstein & Kroese, 2016) to subsample $S_m$ to obtain a dataset that effectively consists of $N \leq m$ i.i.d. samples from $Q$ (here, $N$ is a random variable, but this turns out not to be an issue). Essentially, for each $(x_i, y_i) \in S_m$, we sample a random variable $V_i \sim \text{Uniform}([0, 1])$, and then accept samples where $V_i \leq w^*_{y_i}/b$, where $b$ is an upper bound on $w^*_y$:

$$T_N(S_m, V, w^*, b) = \left\{ (x_i, y_i) \in S_m \ \middle| \ V_i \leq \frac{w^*_{y_i}}{b} \right\}.$$

In our setting, we can take $b = \max_{y \in \mathcal{Y}} w^*_y$. Then, we return $\hat{\tau}(T_N(S_m, V, w^*, b))$. Since $T_N(S_m, V, w^*, b)$ consists of an i.i.d. sample from $Q$, we obtain the desired PAC guarantee (4).

In practice, we do not know the true importance weights $w^*$. Instead, suppose we can obtain intervals $W_y = [\underline{w}^*_y, \overline{w}^*_y]$ such that $w^*_y \in W_y$ with high probability. We let $W = \prod_{y \in \mathcal{Y}} W_y$, and assume $w^* \in W$ with probability at least $1 - \delta$. The algorithm proposed in (Park et al., 2021) adjusts the above algorithm to conservatively account for this uncertainty—i.e., it chooses $\tau$ so the PAC guarantee (4) holds for *any* importance weights $w \in W$:

$$\hat{\tau}(S_m, V, W, b) = \min_{w \in W} \hat{\tau}(T_N(S_m, V, w, b)). \tag{18}$$

We minimize over $\tau$ since choosing smaller $\tau$ leads to larger prediction sets, which is more conservative. (Park et al., 2021) show how to compute (18) efficiently. We have the following guarantee:

**Theorem C.1** (Theorem 4 in (Park et al., 2021)). *Assume that $w^* \in W$. Letting $U = Uniform([0, 1])$,*

$$\mathbb{P}_{S_m \sim P^m, V \sim U^m} \left[ \mathbb{P}_{(X,Y) \sim Q}[y \in C_{\hat{\tau}(S_m, V, W, b)}] \geq 1 - \varepsilon \right] \geq 1 - \delta.$$

In other words, the PAC guarantee (4) holds, with the modification that the outer probability includes the randomness over the samples $V \sim U^m$ used by our algorithm.

## D    ENSURING THE CONFIDENCE BOUNDS AT EACH STEP

The diagonal elements $c_{kk}$ of the confusion matrix of an accurate classifier, are typically much larger than the other elements. Indeed, for an accurate classifier, the probabilities of correct predictions $c_{kk} = P(g(x) = k, y = k)$ are higher than those of incorrect predictions $c_{ik} := P(g(x) = i, y = k), i \neq k$. On the other hand, the Clopper-Pearson interval is expected to be short (for instance, the related Wald interval has length or order $1/\sqrt{m}$, where $m$ is the sample size). Thus, we expect that

$$\overline{c}_{ik}^0 \ll \underline{c}_{kk}^0, k = 1, \dots, K, i \neq k. \tag{19}$$

Without loss of generality, we consider equation 11 as an example. In the Gaussian elimination process, recall that the update at step $t$ is

$$\underline{c}_{ij}^{t+1} = \underline{c}_{ij}^t - \frac{\overline{c}_{ik}^t}{\underline{c}_{kk}^t} \overline{c}_{kj}^t \quad \text{if } i, j > k. \tag{20}$$

Combining with equation 19, the factor by which the $k$-th row is multiplied is small, i.e., $\overline{c}_{ik}^t / \underline{c}_{kk}^t \ll 1$. Thus the resulting $i$-th diagonal elements

$$\underline{c}_{ii}^{t+1} = \underline{c}_{ii}^t - \frac{\overline{c}_{ik}^t}{\underline{c}_{kk}^t} \overline{c}_{ki}^t$$

change little after each elimination step, and are expected to remain positive. Next we discuss intervals for off-diagonal elements.

**Balanced classifier.** For a *balanced classifier*, when $c_{ik}$ and $c_{jk}$ are close for all $i, j$ such that $i \neq k$, $j \neq k$, since the factor $\overline{c}_{ik}^t / \underline{c}_{kk}^t$ is small, the lower bound in equation 20 is expected to be positive.

**Imbalanced classifier.** For the more general case of a possibly imbalanced classifier, $c_{ij}$ and $c_{kj}$ may not be close. This could cause non-positive bounds at certain steps, so the confidence interval may not be valid at the next steps; e.g., equation 13 may fail. However, note that since

$$c_{ik}^t \in [\underline{c}_{ik}^t, \overline{c}_{ik}^t], \quad c_{kj}^t \in [\underline{c}_{kj}^t, \overline{c}_{kj}^t],$$

we have

$$c_{ik}^t c_{kj}^t \leq \max\{|\underline{c}_{ik}^t|, |\overline{c}_{ik}^t|\} \cdot \max\{|\underline{c}_{kj}^t|, |\overline{c}_{kj}^t|\}$$

and hence

$$\frac{c_{ik}^t c_{kj}^t}{c_{kk}^t} \leq \frac{\max\{|\underline{c}_{ik}^t|, |\overline{c}_{ik}^t|\} \cdot \max\{|\underline{c}_{kj}^t|, |\overline{c}_{kj}^t|\}}{\underline{c}_{kk}^t}.$$

In fact, one can derive the even tighter bound

$$\max\left( \frac{\overline{c}_{ik}^t \overline{c}_{kj}^t}{\underline{c}_{kk}^t}, \frac{\underline{c}_{ik}^t \overline{c}_{kj}^t}{\underline{c}_{kk}^t}, \frac{\underline{c}_{ik}^t \underline{c}_{kj}^t}{\underline{c}_{kk}^t}, \frac{\overline{c}_{ik}^t \underline{c}_{kj}^t}{\underline{c}_{kk}^t}, \frac{\overline{c}_{ik}^t \overline{c}_{kj}^t}{\overline{c}_{kk}^t}, \frac{\underline{c}_{ik}^t \overline{c}_{kj}^t}{\overline{c}_{kk}^t}, \frac{\underline{c}_{ik}^t \underline{c}_{kj}^t}{\overline{c}_{kk}^t}, \frac{\overline{c}_{ik}^t \underline{c}_{kj}^t}{\overline{c}_{kk}^t} \right).$$

This can be checked by carefully going through all possible cases of positive and negative values of the bounds. Similar changes can be made to computing the upper bounds.

It is possible for our final interval $W$ to contain negative lower bounds due to loose element-wise intervals or other factors. Since importance weights are non-negative, negative importance weight bounds act the same way as zero lower bounds in rejection sampling, and preserve our guarantees.

Finally, the requirement of an accurate classifier is already imposed by methods such as BBSE to ensure the invertibility of the confusion matrix. Therefore, our Gaussian elimination approach does not impose significantly stronger assumptions.

## E    PROOF OF LEMMA 3.1

For the first phase, we prove by induction on $t$ that (10) holds for all $t$. The base case $t = 0$ holds by assumption. For the induction step, we focus on $\underline{c}_{ij}^{t+1}$; the remaining bounds $\overline{c}_{ij}^{t+1}$, $\underline{q}_k^{t+1}$, and $\overline{q}_k^{t+1}$ follow similarly. There are three sub-cases i, each corresponding to one of the update rules in (11). For the first update rule $\overline{c}_{ij}^{t+1} = 0$, equation 10 follows since the Gaussian elimination algorithm

guarantees that $c_{ij}^{t+1} = 0$. For the second and third update rules, equation 10 follows by direct algebra and the induction hypothesis. For instance, for $i, j > t$, equation 13 holds, and similarly $c_{ij}^{t+1} \geq \underline{c}_{ij}^{t+1}$.

For the second phase, the fact that $\underline{s} \leq s \leq \overline{s}$ and $\underline{w}^* \leq w^* \leq \overline{w}^*$ follows by a similar induction argument. Since Gaussian elimination guarantees that $w^* = \mathbf{C}_P^{-1} q^*$, and we have shown that $w^* \in W = \prod_{i=1}^{K} [\underline{w}_i, \overline{w}_i]$, the claim follows. $\square$

## F PROOF SKETCH OF THEOREM 3.2

First, Theorem 3.2 follows from equation 5, assuming the given confidence intervals $W_i = [\underline{w}_i, \overline{w}_i]$ for each importance weight $w_i^*$ is valid—i.e., $w_i^* \in W_i$. Equation 5 follows by Theorem 4 in Park et al. (2021). Roughly speaking, if $w_i^*$ is known, then we can use a standard rejection sampling procedure based on $w_i^*$ to convert $S_m$ into a set of i.i.d. samples from $Q$. Then, the PAC guarantee would follow by standard conformal prediction results, e.g., Park et al. (2020). When $w_i^*$ is not known, then intuitively, Park et al. (2021) takes the worst case over all $w_i^* \in W_i$. They use a reparameterization trick to do so in a computationally efficient way. Finally, by Lemma 3.1, we have $w_i^* \in W_i$ for all $i \in [n]$ with high probability, so Theorem 3.2 follows by a union bound.

## G CONSERVATIVE BASELINE

We describe PS-C, the conservative baseline summarized in Algorithm 2. In particular, given an upper bound $b \geq w^*$ on the importance weight, we use the upper bound

$$\mathbb{E}_{(X,Y) \sim P}[\ell(g(X), Y) \cdot w_Y^*] \leq b \cdot \mathbb{E}_{(X,Y) \sim P}[\ell(g(X), Y)].$$

As a consequence, we can run the original prediction set algorithm from Vovk (2012); Park et al. (2019) with a more conservative choice of $\varepsilon$ that accounts for this upper bound.

---

**Algorithm 2** PS-C: an algorithm using the CP bound in equation 21.

---

1: **procedure** PS-C($S_m, T_n^X, f, \mathcal{T}, \varepsilon, \delta_w, \delta_C$)
2: $\quad \underline{c}, \overline{c}, \underline{q}, \overline{q} \leftarrow \text{CPINTERVAL}(S_m, T_n^X, x \mapsto \arg\max_{y \in \mathcal{Y}} f(x, y), \delta_w)$
3: $\quad W \leftarrow \text{INTERVALGAUSSIANELIM}(\underline{c}, \overline{c}, \underline{q}, \overline{q})$
4: $\quad b \leftarrow \max_{k \in [K]} \overline{w}_k$
5: $\quad$ **return** PS($S_m, f, \mathcal{T}, \varepsilon/b, \delta_C$)

---

**Lemma G.1.** *Algorithm 2 satisfies the PAC guarantee under label shift equation 4.*

*Proof.* Having constructed the importance weight intervals $w^*$, we can use $b = \max_{k \in [K]} \overline{w}_k$ to find a conservative upper bound on the risk as follows:

$$E_{(X,Y) \sim Q}[\mathbb{1}(Y \notin C_\tau(X))] = \int q(x, y) \mathbb{1}(y \notin C_\tau(x)) dx dy$$

$$= \int p(x, y) w(y) \mathbb{1}(y \notin C_\tau(x)) dx dy \leq b E_{(X,Y) \sim P}[\mathbb{1}(Y \notin C_\tau(X))]. \tag{21}$$

Hence, using the PS prediction set algorithm with parameters $(\varepsilon/b, \delta)$, the output is $(\varepsilon, \delta)$-PAC. $\square$

## H ORACLE IMPORTANCE WEIGHT RESULTS

Here, we show comparisons to an oracle that is given the ground truth importance weights (which are unknown and must be estimated in most practical applications). It uses rejection sampling according to these weights rather than conservatively accounting for uncertainty in the weights. In contrast to our baselines (which need to estimate $w$), this oracle represents a "gold (yet not achievable) standard" to compare with. It enables us to quantify the increase in average prediction set size due to uncertainty in our estimates of $w$, stemming from (i) finite sample error, accounted for by Clopper-Pearson intervals, and (ii) label shift, accounted for by our interval Gaussian elimination algorithm.

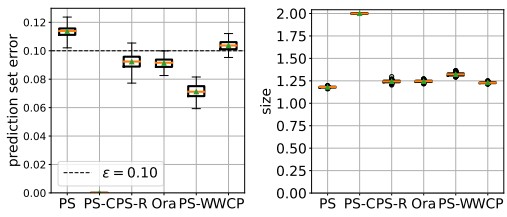 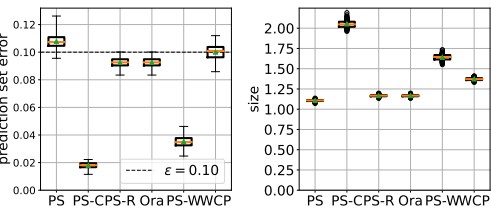

(a) Prediction set error and size on the CDC dataset. Parameters are $\varepsilon = 0.1$, $\delta = 5 \times 10^{-4}$, $m = 42000$, $n = 42000$, and $o = 9750$.

(b) Prediction set error and size on the CIFAR10. Parameters are $\varepsilon = 0.1$, $\delta = 5 \times 10^{-4}$, $m = 27000$, $n = 19997$, and $t = 19997$.

Figure 5: Prediction set results with comparison to the oracle importance weight (Ora).

First, for the CDC heart dataset, we consider the following label distributions: source $(94\%, 6\%)$, target: $(63.6\%, 36.4\%)$; see Figure 5a for the results. Second, for the CIFAR-10 dataset, we consider the following label distributions: source $(10\%, 10\%, 10\%, 10\%, 10\%, 10\%, 10\%, 10\%, 10\%, 10\%)$, target: $(6.7\%, 6.7\%, 6.7\%, 40.0\%, 6.7\%, 6.7\%, 6.7\%, 6.7\%, 6.7\%, 6.7\%)$; see Figure 5b for results.

# I ADDITIONAL RESULTS

## I.1 CDC HEART

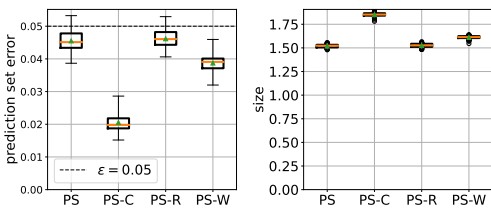 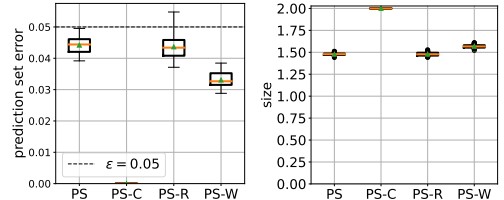

(a) Prediction set error and size under small shifts on the CDC Heart dataset. Parameters are $\varepsilon = 0.05$, $\delta = 5 \times 10^{-4}$, $m = 42000$, $n = 42000$, and $o = 9750$.

(b) Prediction set error and size under large shifts on the CDC Heart dataset, Parameters are $\varepsilon = 0.05$, $\delta = 5 \times 10^{-4}$, $m = 42000$, $n = 42000$, and $o = 9750$.

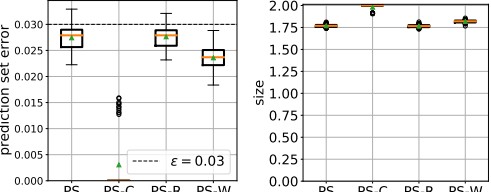 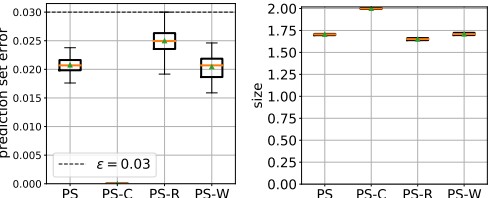

(c) Prediction set error and size under small shifts on the CDC Heart dataset. Parameters are $\varepsilon = 0.03$, $\delta = 5 \times 10^{-4}$, $m = 42000$, $n = 42000$, and $o = 9750$.

(d) Prediction set error and size under large shifts on the CDC Heart dataset, Parameters are $\varepsilon = 0.03$, $\delta = 5 \times 10^{-4}$, $m = 42000$, $n = 42000$, and $o = 9750$.

Figure 6: More Prediction set results with different hyperparameters on the CDC Heart dataset.

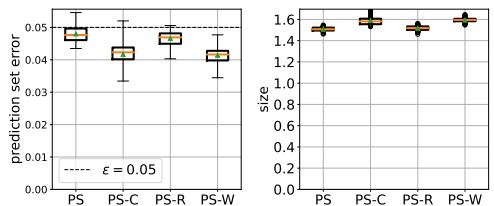 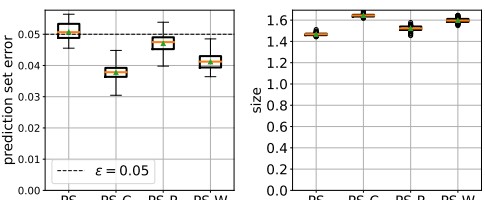

(a) Prediction set error and size under *reversed small* shifts $((91.3\%, 8.7\%) \rightarrow (94\%, 6\%))$ on the CDC Heart dataset. Parameters are $\varepsilon = 0.05$, $\delta = 5 \times 10^{-4}$, $m = 42000$, $n = 42000$, and $o = 9750$.

(b) Prediction set error and size under *reversed large* shifts $((63.6\%, 36.4\%) \rightarrow (94\%, 6\%))$ on the CDC Heart dataset, Parameters are $\varepsilon = 0.05$, $\delta = 5 \times 10^{-4}$, $m = 42000$, $n = 42000$, and $o = 9750$.

Figure 7: More Prediction set results with different hyperparameters on the CDC Heart dataset.

## I.2 CIFAR-10

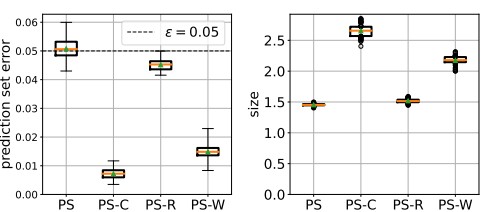 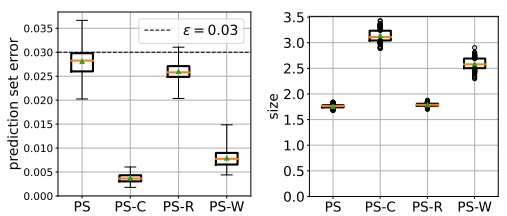

(a) Prediction set error and size with larger shift on the CIFAR-10. Parameters are $\varepsilon = 0.05$, $\delta = 5 \times 10^{-4}$, $m = 27000$, $n = 19997$, and $t = 19997$.

(b) Prediction set error and size with larger shift on the CIFAR-10. Parameters are $\varepsilon = 0.03$, $\delta = 5 \times 10^{-4}$, $m = 27000$, $n = 19997$, and $t = 19997$.

Figure 8: More Prediction set results with different hyperparameters on the CIFAR-10 dataset.

## I.3 AGNEWS

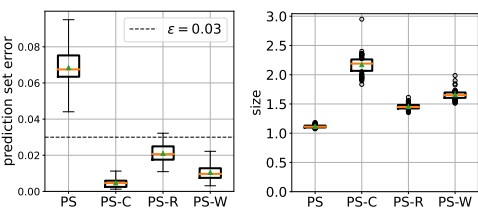 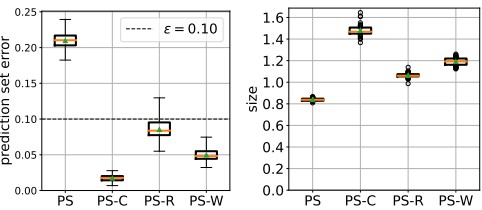

(a) Prediction set error and size on the AGNews Dataset. Parameters are $\varepsilon = 0.03$, $\delta = 5 \times 10^{-4}$, $m = 26000$, $n = 12800$, and $t = 12800$.

(b) Prediction set error and size on the AGNews Dataset. Parameters are $\varepsilon = 0.1$, $\delta = 5 \times 10^{-4}$, $m = 26000$, $n = 12800$, and $t = 12800$.

Figure 9: More Prediction set results with different hyperparameters on the AGNews dataset.

## I.4 CHESTX-RAY

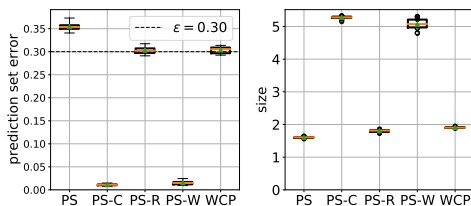 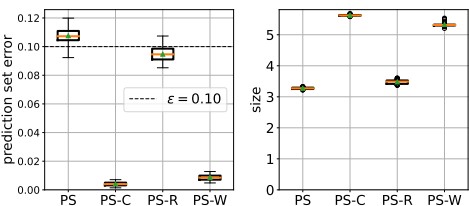

(a) Prediction set error and size on the ChestX-ray dataset. Parameters are $\varepsilon = 0.3$, $\delta = 5 \times 10^{-4}$, $m = 33600$, $n = 17600$, and $t = 3520$.

(b) Prediction set error and size on the ChestX-ray dataset. Parameters are $\varepsilon = 0.1$, $\delta = 5 \times 10^{-4}$, $m = 67200$, $n = 35200$, and $t = 3520$.

Figure 10: More Prediction set results with different hyperparameters on the ChestX-ray dataset.

### I.5 ADDITIONAL BASELINES

Class-conditional conformal predictors fit separate thresholds for each label and demonstrate robustness to label shift. In Figure 11, we show the class-conditional results for both conformal prediction tuned for average coverage and our PAC prediction set, on the CDC and CIFAR-10 dataset. Here, LWCP is a baseline from the label-conditional setting from Sadinle et al. (2019), which does not satisfy a PAC guarantee. PS-LW adapts the standard PAC prediction set algorithm (Vovk, 2012; Park et al., 2020) to the label-conditional setting; our approach is PS-W. Most relevantly, while PS-LW approximately satisfies the desired error guarantee, it is more conservative than our approach (PS-W) and produces prediction sets that are larger on average. Intuitively, it satisfies a stronger guarantee than necessary for our setting, leading it to be overly conservative.

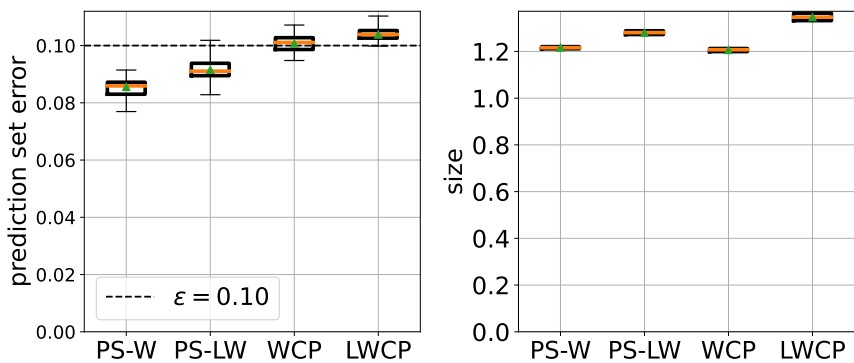

Figure 11: Prediction set error and size on the CDC dataset under small shifts. Parameters are $\varepsilon = 0.1$, $\delta = 5 \times 10^{-4}$, $m = 42000$, $n = 42000$, and $o = 9750$.

Empirically, we find that while APS improves coverage in the label shift setting, it does not satisfy our desired PAC guarantee. In particular, we show results for the APS scoring function with vanilla prediction sets in Figure 12; as can be seen, it does not satisfy the desired coverage guarantee. Due to its unusual structure, it is not clear how APS can be adapted to the PAC setting, which is our focus.

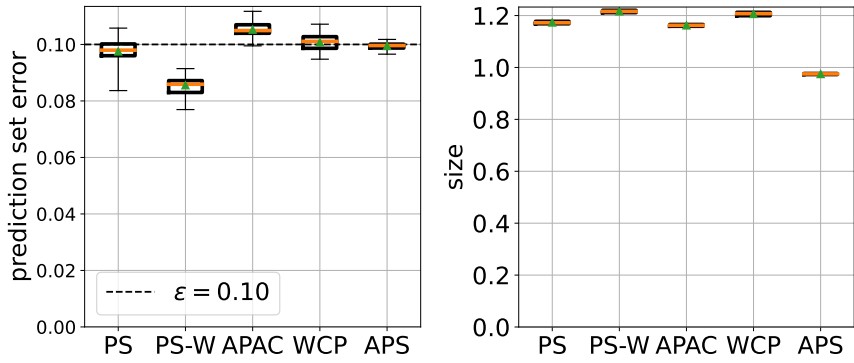

Figure 12: Prediction set error and size on the CDC dataset. Parameters are $\varepsilon = 0.1$, $\delta = 5 \times 10^{-4}$, $m = 42000$, $n = 42000$, and $o = 9750$. Label distribution $(94\%, 6\%)$ with small shift $(91.3\%, 8.7\%)$ in the target.

### I.6 CIFAR-100

We show results on CIFAR-100 in Figure 13. The scores are the logits of a pretrained ViT model. The results show that both PS-C and PS-W attain the desired coverage guarantee, while the remaining approaches do not do so. Furthermore, PS-W outperforms PS-C in terms of average prediction set size.

In this case, both PS-C and PS-W are quite conservative, due to two main reasons. First, the Clopper-Pearson intervals can be conservative when $\delta$ is very small, and we need to divide $\delta$ by $K(K+1)+1$, since we need to take a union bound over $K(K+1)+1$ events in equation 8. Second, although our Gaussian "interval" elimination algorithm does uncertainty propagation in time $O(K^3)$, the resulting confidence intervals $(\underline{w}, \overline{w})$ may be conservative, thereby amplifying the first issue. With more calibration data, our approach would be less conservative.

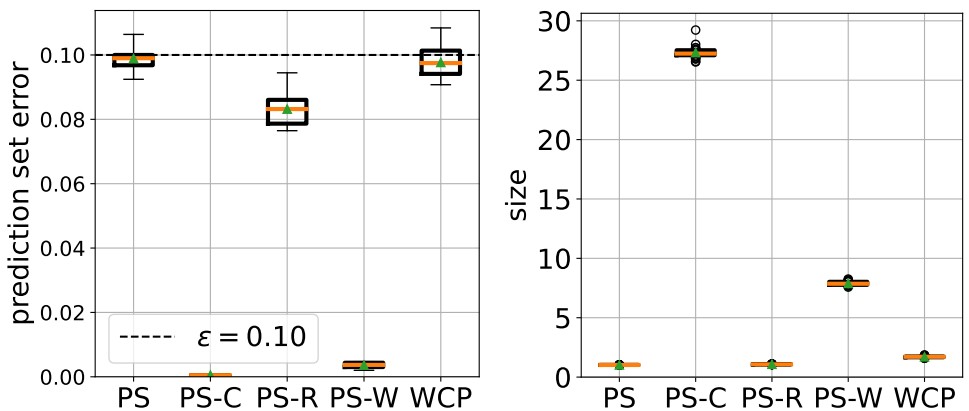

Figure 13: Prediction set error and size on the CIFAR-100 dataset. Parameters are $\varepsilon = 0.1$, $\delta = 5 \times 10^{-4}$, $m = 270k$, $n = 180k$, and $o = 5950$. Label distribution is $([1.01\%] \times 99 + [0.3\%])$ for source, and $([0.84\%] \times 99 + [16.8\%])$ for target.

## I.7 LARGE $\varepsilon$

We provide results on larger choices of $\varepsilon$ in Figures 14 & 15.

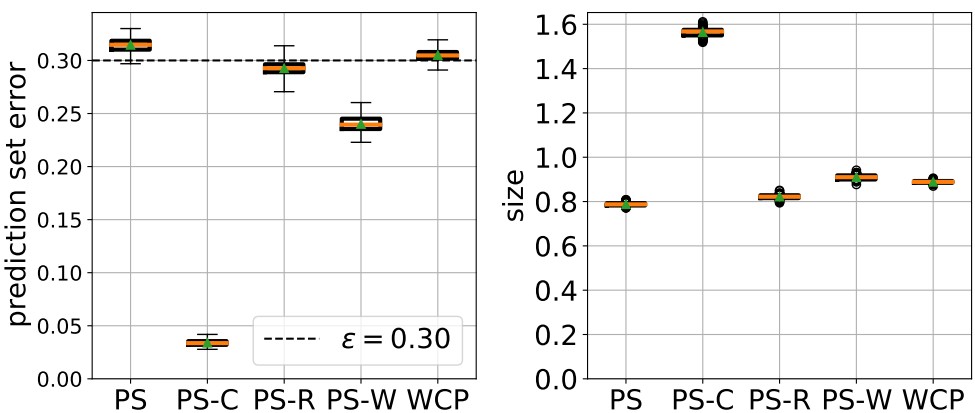

Figure 14: Prediction set error and size on the CDC dataset. Parameters are $\varepsilon = 0.3$, $\delta = 5 \times 10^{-4}$, $m = 42000$, $n = 42000$, and $o = 9750$. Label distribution is $(94\%, 6\%)$ for source, and $(63.6\%, 36.4\%)$ for target.

## I.8 NON-UNIFORM SOURCE DISTRIBUTIONS

We provide results with a non-uniform source distribution in Figure 16.

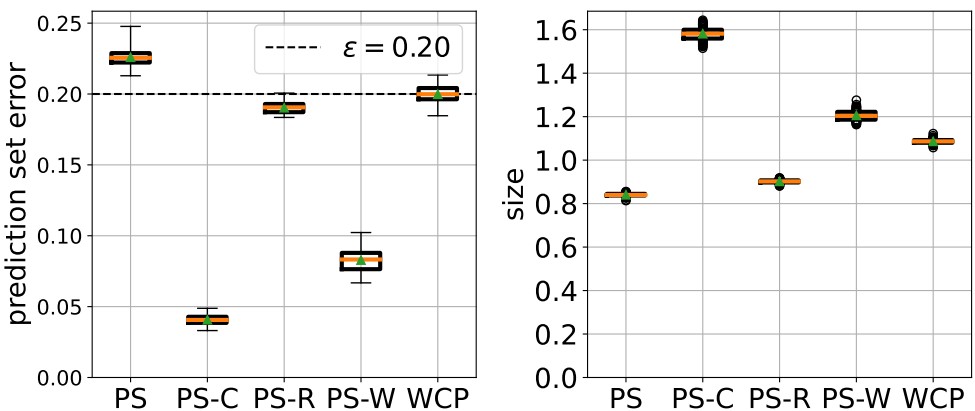

Figure 15: Prediction set error and size with large shifts on the CIFAR-10 dataset. Parameters are $\varepsilon = 0.2$, $\delta = 5 \times 10^{-4}$, $m = 27000$, $n = 19997$, and $t = 19997$.

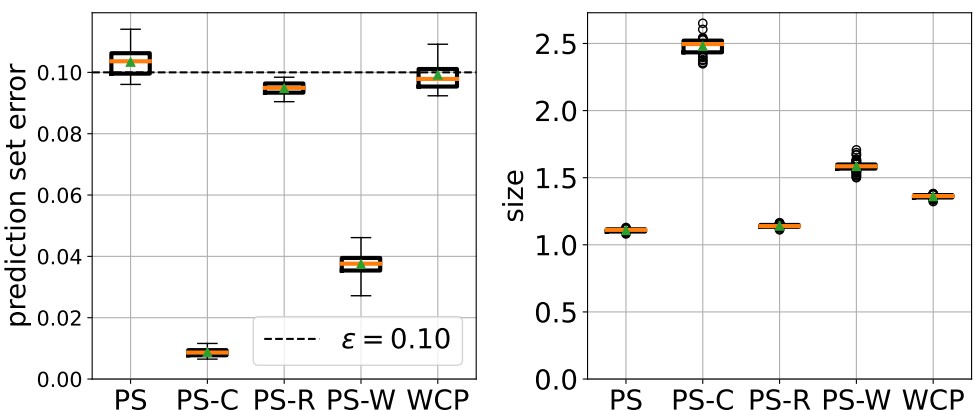

Figure 16: Prediction set error and size on the CIFAR-10 dataset. Parameters are $\varepsilon = 0.1$, $\delta = 5 \times 10^{-4}$, $m = 34k$, $n = 45k$, and $o = 45k$. Label distribution is $(10.53\%, 5.26\%, 15.79\%, 5.26\%, 5.26\%, 10.53\%, 15.79\%, 10.53\%, 5.26\%, 15.79\%)$ for source, and $(3.57\%, 7.14\%, 10.71\%, 35.71\%, 3.57\%, 7.14\%, 10.71\%, 3.57\%, 7.14\%, 10.71\%)$ for target.

