# OpenReview forum: "PAC Prediction Sets Under Label Shift"
_ICLR.cc/2024/Conference — ICLR 2024 poster_

### Official Review · Reviewer_tiZb · 2023-10-31

**Soundness:** 3 good
**Presentation:** 3 good
**Contribution:** 3 good
**Rating:** 8
**Confidence:** 4

**Summary:**

The paper introduces an innovative algorithm for the construction of prediction sets that promises probably approximately correct (PAC) guarantees in scenarios with label shift. Such prediction sets provide a mechanism to capture uncertainties by predicting sets of labels instead of individual ones. The approach seeks to address the challenges posed by distribution shifts and applies importance weighting techniques to overcome these challenges, coupled with the novel introduction of Gaussian elimination for uncertainty propagation. The approach is validated empirically across a mix of datasets spanning multiple domains.

**Strengths:**

Label shift is an inherent challenge in real-world datasets, especially in settings like healthcare where outcomes can dynamically change. Addressing uncertainty under such scenarios is a meaningful contribution.

The paper presents a fresh take on dealing with label shift by estimating importance weights and subsequently leveraging Gaussian elimination. The use of Gaussian elimination to propagate uncertainties in the domain of prediction set construction is notable and potentially groundbreaking.

The empirical evaluation covers a variety of datasets from distinct domains, lending more weight and general applicability to the findings.

The example provided for the ChestX-ray dataset effectively illustrates the applicability and significance of the approach in practical scenarios.

**Weaknesses:**

While the methodology is fundamentally sound, it might benefit from a more in-depth discussion or a visual representation to aid clarity, especially around the estimation and application of importance weights.

It would be helpful to know the performance of the proposed method in relation to existing approaches in more detail, especially to understand the trade-offs or advantages.

It's unclear how the method would perform under extreme label shifts or in scenarios with limited data availability.

**Questions:**

How robust is the algorithm to extreme label shifts or when the shift is non-uniform across classes?
How sensitive is the approach to the estimation errors in the importance weights, and are there any mechanisms to mitigate potential inaccuracies?
Are there computational overheads introduced by the Gaussian elimination procedure, especially when applied to large datasets?

---

> ### Author Response · Authors · 2023-11-18
>
> 1. It would be helpful to know the performance of the proposed method in relation to existing approaches in more detail, especially to understand the trade-offs or advantages.
>
>     A: At a high level, our baselines fall into two categories: i) Approaches that do not provide a PAC guarantee under label shift, and therefore do not satisfy our PAC coverage guarantee. In particular, PS, PS-R, and WCP fall into this category. Note that PS-R accounts for label shift but not in a rigorous way, since it ignores uncertainty in the importance weights. WCP also accounts for label shift; however, it provides a weaker coverage guarantee so it also does not satisfy our PAC coverage guarantee.
>
>     ii) Algorithms designed to satisfy a PAC coverage guarantee under label shift, but do so in a na\"{i}ve way that is overly conservative. PS-C falls into this category; as can be seen from our experiments, it satisfies our PAC coverage guarantee, but its average prediction set size is significantly larger than ours.
>
>     Thus, the tradeoff is between satisfying our PAC coverage guarantee under label shift, and constructing small prediction sets. Our algorithm optimizes this tradeoff: it constructs the smallest prediction sets among algorithms that satisfy the PAC coverage guarantee.
>
> 2. Robustness under non-uniform distribution across classes?
>
>     A: Both theoretically and empirically, our algorithm and PAC guarantee apply to non-uniform label distributions. We show results for CIFAR-10, resampled to achieve a non-uniform source and target distribution, in Section I.8 of the revision. In our paper, our results for CDC Heart and AGNews have a non-uniform source distribution.
>
> 3. Robustness under extreme label shifts or in scenarios with limited data availability.
>
>     A: In the paper, we show results for very small shifts for both CIFAR-10 and CDC Heart datasets. For extremely large shifts, our approach might produce very conservative results, since the imbalance in the confusion matrix may lead to more conservative relaxed intervals for each element, thereby affecting each step.
>
>     Similarly, limited data availability would result in relaxed bounds for the Clopper-Pearson intervals under our setting of a very small $\delta$. We note that these issues would affect all algorithms including our baselines, since the problem becomes intrinsically much more difficult. We've added this to the revision.
>
> 4. How sensitive is the approach to the estimation errors in the importance weights, and are there any mechanisms to mitigate potential inaccuracies?
>
>     A: Importantly, our confidence intervals are guaranteed to contain the true importance weights with high probability (Lemma 3.1), and both the estimation error and the failure probability are accounted for by our algorithm. Thus, our algorithm fully accounts for potential inaccuracies in the importance weights.
>
> 5. Are there computational overheads introduced by the Gaussian elimination procedure, especially when applied to large datasets?
>
>     A: Our novel interval Gaussian elimination algorithm (our main technical contribution) is as efficient as traditional Gaussian elimination up to constant factors, with running time $O(K^3)$ (and the constant is relatively small). In practice, we find that its running time is negligible compared to running the model, demonstrating the effectiveness of our technique.

---

> > ### Comment · Reviewer_tiZb · 2023-11-22
> >
> > Thank you for the clarification. I maintain my high score.

---

### Official Review · Reviewer_2Pyj · 2023-11-01

**Soundness:** 3 good
**Presentation:** 3 good
**Contribution:** 3 good
**Rating:** 6
**Confidence:** 3

**Summary:**

The paper introduces a novel algorithm for constructing prediction sets with PAC (probably approximately correct) guarantees under the label shift setting. The proposed method leverages the concept of importance weighting, drawing upon the confusion matrix and the distribution of predicted labels. The authors construct confidence intervals for these elements and develop an algorithm to propagate these intervals using Gaussian elimination. Through a rejection sampling-based approach (Park et al. 2022), they attain prediction sets with PAC guarantees.

**Strengths:**

- The paper offers a method for uncertainty quantification in machine learning under label shift assumption.
- The paper mostly relies mostly on previous work, the innovation is propagating confidence intervals through Gaussian elimination to quantify the uncertainty of weights.
- The paper is overall well-written and organized.

**Weaknesses:**

- While a theoretical guarantee is provided, it is dependent on $\tau$, which seems less meaningful.
- The assumption that the confusion matrix is invertible might be too strong for real-world applications. The element-wise confidence intervals approach for importance weights can be computationally intensive, especially given the large number of iterations required.
- The simulations could potentially be strengthened (see questions below).

**Questions:**

- On Page 3: $p(\hat{y}|y) = P_{(X,Y)\sim P_X} \[g(X) = \hat{y}| Y = y \]$. Should it be $P_{(X,Y)\sim P}$, since $g$ is trained on $X$ and $Y$.

In empirical studies:
- On Page 7: "We select $\epsilon$ for each dataset in a way that the resulting average prediction set size is greater than one." This approach seems to deviate from the typical use of conformal prediction where any desired coverage level can be achieved.
- According to (Guan and Tibshirani, 2022), a prediction $C(x) = \emptyset$ would suggest that x is likely distant from the training data. This implies that we cannot assign it to any class and should consider it an outlier. I'm curious if the proposed method can serve a similar purpose in outlier detection.
- On Pages 8-9: Why are the sample sizes for the CIFAR-10 and AGNews experiments different between the "large shift" and "small shift" scenarios?
- How did you determine the number of iterations, $t$? In section 3, it seems $t$ should be $\[K-1\]$. However, in the empirical studies,
$t$ appears to range from 10,000 to 20,000, even for binary classification problems. Also, does this imply a significant computational cost?
- In Figure 5, you've replaced "PS-W" (the proposed method) with "Ora". I'm curious, did you use the oracle weight for the other methods as well?

---

> ### Author Response · Authors · 2023-11-18
>
> 1. While a theoretical guarantee is provided, it is dependent on $\tau$, which seems less meaningful.
>
>     A: Apologies for the misunderstanding, but the guarantee does not  depend on $\tau$. Or, to be precise, $\tau$ is not a hyperparameter; instead, it is a threshold parameter that we estimate based on the calibration dataset. Thus, our guarantee is satisfied for any user-provided $(\epsilon, \delta)$, and does not depend on $\tau$.
>
> 2. The assumption that the confusion matrix is invertible might be too strong for real-world applications.
>
>     A: First, we emphasize that the assumption of invertibility of the confusion matrix is well-accepted in the label shift setting (Lipton 2018); in particular, we have not introduced any new assumptions. Second, as described in the Lipton (2018), this assumption is reasonable as long the classifier obtains high predictive accuracy, which is true of modern classifiers even with dependent classes (e.g., ImageNet). Finally, our algorithm checks that this assumption holds; if it does not hold, it can notify the user that our label shift technique is not applicable. Thus, we believe this assumption is reasonable in many real world applications.
>
> 3. The element-wise confidence intervals approach for importance weights can be computationally intensive, especially given the large number of iterations required.
>
>     A: The number of iterations is bounded by the number of classes $K$. Specifically, the Gaussian interval elimination method involves fixed polynomial time computation $O(K^3)$.
>
>     We apologize for potential misunderstanding caused by the overloaded use of $t$. In Gaussian elimination, $t$ refers to a step. In our experiments, $t$ was used to denote the test set size. We have fixed this in our revision.
>
> 4. On Page 3: $p(\hat{y}|y) = P_{(X,Y)\sim P_X}[g(X)=\hat{y}|Y=y]$. Should it be $P_{(X,Y)\sim P}$, since $g$ is trained on $X$ and $Y$.
>
>     A: Thank you for pointing this out, we have fixed this in the revision.
>
> 5. "Selection of $\epsilon$ in a way that the resulting average prediction set size is greater than one." seems to deviate from the typical use of conformal prediction where any desired coverage level can be achieved.
>
>     A: Indeed, our algorithm and guarantees hold for any choice of $\epsilon \in [0,1]$. The reason we chose $\epsilon$ in this way was to focus on settings where the prediction sets have nontrivial sizes.
>
>     When the prediction set sizes are always one, the algorithm can simply include only the top label in the set and still guarantee coverage, which is a trivial form of uncertainty quantification. Our goal is to focus on settings where the algorithm must perform some nontrivial uncertainty quantification to guarantee coverage.
>
>     For completeness, we include results for both the CDC Heart and CIFAR-10 datasets in Section I.7, where the average prediction set size is equal to (or even less than) one. As can be seen, our algorithm continues to guarantee coverage in these settings.

---

> ### Author Response · Authors · 2023-11-18
>
> 6. According to (Guan and Tibshirani, 2022), a prediction $C(x)=\emptyset $ would suggest that $x$ is likely distant from the training data. I'm curious if the proposed method can serve a similar purpose in outlier detection.
>
>     A: We have carefully read the references provided by the reviewer; they focus specifically on the out-of-distribution (OOD) setting, and study a risk control problem distinct from our PAC coverage guarantees.
>
>     As noted in prior work [1,2], empty prediction sets can be interpreted as having too high of an uncertainty. Split conformal prediction chooses a threshold $\tau$ and includes all labels for which the scoring function $f$ exceeds $\tau$---i.e., $\tilde{f}(x)=${$y\mid f(x,y)\ge \tau$}. As a consequence, if $f(x,y)$ is near-uniform, then none of of the labels may satisfy this condition and the prediction set is empty.
>
>     As a consequence, our approach could be used as a heuristic for detecting examples $x$ that are far from the training distribution. However, there are two important caveats to keep in mind. First, examples far from the training data can have highly confident predictions (as demonstrated by the existence of high-confidence adversarial examples), which would result in non-empty prediction sets. Second, there can be other explanations for large prediction sets, such as a high amount of label noise (i.e., $x$ may correspond to many labels $y$, even in the ground truth). For instance, this might be true of medical prediction tasks, where there is a high amount of intrinsic uncertainty in patient outcomes.
>
>     [1] Guan, Leying, and Robert Tibshirani. "Prediction and outlier detection in classification problems." Journal of the Royal Statistical Society Series B: Statistical Methodology 84.2 (2022): 524-546.
>
>     [2] Han, Yujin, Mingwenchan Xu, and Leying Guan. "Conformalized semi-supervised random forest for classification and abnormality detection." arXiv preprint arXiv:2302.02237 (2023).
>
> 7. Why are the sample sizes for the CIFAR-10 and AGNews experiments different between the "large shift" and "small shift" scenarios?
>
>     A: The size difference is primarily caused by the label shift, as each label has a different proportion. Indeed, we did not intentionally adjust the size.
>
> 8. In Figure 5, you've replaced "PS-W" (the proposed method) with "Ora". I'm curious, did you use the oracle weight for the other methods as well?
>
>     A: We only used the oracle weight in the "Ora" approach; all of our baselines estimate the weights $w$ (if they are needed). The main point of showing the "Ora" results is to quantify how much our algorithm "loses" by needing to estimate the weights instead of knowing them exactly. We've clarified this in the revision.

---

> ### Comment · Reviewer_2Pyj · 2023-11-22
>
> Thanks for your detailed responses. They've addressed my concerns well, so I'm raising my score.

---

### Official Review · Reviewer_onAF · 2023-11-01

**Soundness:** 3 good
**Presentation:** 3 good
**Contribution:** 2 fair
**Rating:** 6
**Confidence:** 3

**Summary:**

The authors consider the conformal prediction (CP) interval for the classification problem under the label shift setting, where the distribution of the labels may shift, but the distribution of covariates conditioned on the labels remains fixed. Following the estimation of the framework from previous work, the authors take into account the uncertainty in estimating the confusion matrix and distribution of predicted labels when constructing the prediction interval.

**Strengths:**

The paper is well-explained and easy to follow.

**Weaknesses:**

see below.

**Questions:**

1. When comparing the work of Sadinle (LWCP), the coverage is not guaranteed. However, in my perspective, it seems to be more proper comparing the proposed method to the class-specific construction which allows different cut-off for different classes and guarantees the coverage under arbitrary shift.
2. Do you mean WCP instead of CP in Figure 2?

---

> ### Author Response · Authors · 2023-11-18
>
> 1. When comparing the work of Sadinle (LWCP), the coverage is not guaranteed. However, in my perspective, it seems to be more proper comparing the proposed method to the class-specific construction which allows different cut-off for different classes and guarantees the coverage under arbitrary shift.
>
>     A: First, traditional conformal prediction algorithms, including LWCP, are not guaranteed to satisfy our PAC guarantee (instead, they satisfy a weaker guarantee). In addition, we compare to PS-LW in Appendix H.5, which can be viewed as an adaptation of LWCP to the PAC setting; as can be seen, it tends to produce larger prediction sets than our approach.
>
> 2. WCP instead of CP in Figure 2?
>
>     A: Yes, sorry for the typo; we have fixed it in our revision.

---

> ### Comment · Reviewer_onAF · 2023-11-21
>
> I don't agree with the author that traditional algorithms do not guarantee coverage under label shifts (depending on which one you are looking at). As I mentioned, in the original paper of Sadinle, setting alpha_y = alpha in the class-specific coverage provides coverage on the test samples under an arbitrary label shift at the desired coverage 1-alpha. (This per-class coverage has also been used in [Guan and Tibshirani, 2022] (reference [1] in the authors' reply to Reviewer 2Pyj), which aims for label coverage under arbitrary shifts while minimizing the average set length on the test samples.)
>
> Hence, such per-class coverage guarantees are not weaker than the proposed method unless I understand the coverage guarantee of PSW incorrectly (please feel free to correct me if so), and that's why I recommend including comparisons with the class-specific version of Sadinle. It reads strange to me when one version of the referenced work provides the desired theoretical guarantee but is left out of the comparisons.

---

> > ### Author Response · Authors · 2023-11-22
> >
> > Thank you for taking the time to clarify your question! To clarify our response, we have already performed the relevant comparisons:
> >
> > 1) Traditional, *non-PAC* label-conditional conformal prediction algorithms such as Sadinle et al. guarantee coverage, but do not guarantee *PAC* coverage. This is the "LWCP" (for label-wise conformal prediction) baseline in Figure 11. As can be seen, it satisfies the traditional, *non-PAC* coverage guarantee on average over random calibration datasets, whereas a PAC coverage guarantee satisfies it with high probability over the calibration dataset.
> >
> > 2) We can adapt label-conditional algorithms to the PAC setting by using a different threshold per class. This is the "PS-LW" (for prediction set -- label-wise) baseline in Figure 11. As can be seen, it satisfies the PAC coverage guarantee, but it is more conservative than our algorithm. This is because satisfying the coverage guarantee individually for each label is more difficult than satisfying it for a specific label shift.
> >
> > To the best of our understanding, 2. is exactly the baseline you are suggesting. Please let us know if you have any additional questions!

---

> > > ### Comment · Reviewer_onAF · 2023-11-22
> > >
> > > This is the results from Sadinle's paper, section 2.2, for class-specific/label-conditional classifier:
> > >
> > > Theorem 2 Given a set of error levels {αy}Ky = 1, the set-valued classifier induced by the sets Cy = {x: p(y|x) ⩾ ty}, with ty chosen so that ℙ(Cy∣∣Y=y)=1−αy, simultaneously minimizes the probabilities of incorrect label assignments for all y and the ambiguity.
> > >
> > > How can it not guarantee coverage under label shift when setting alpha_y = alpha? It guarantees class-wise coverage and allows for an arbitrary mixture of labels in the test samples.
> > >
> > > I am willing to raise my score if this is my misunderstanding or if this is resolved by the authors.

---

> > > > ### Author Response · Authors · 2023-11-22
> > > >
> > > > You are correct that Saldinle et al. provides a **marginal** coverage guarantee under the label shift setting. However, it does not provide a **PAC** coverage guarantee, which is stronger than the marginal coverage guarantee traditionally provided by conformal prediction. To be specific, Saldinle et al. provides a marginal coverage guarantee of the form
> > > >
> > > > $$\mathbb{P}_{S \sim P^n,(x,y^*) \sim P}[y^* \in C_S(x)] \ge \alpha$$
> > > >
> > > > However, the goal of our paper is to obtain a stronger $\(\epsilon, \delta\)$-guarantee (i.e. PAC guarantee):
> > > >
> > > > $$P_{S\sim P^n}[P_{(x,y^*)\sim P}[y^*\in C_S(x)]\ge1-\epsilon]\ge1-\delta$$
> > > >
> > > > In particular, the latter guarantee separately bounds the coverage rate by $1-\epsilon$ and the failure rate due to the randomness in the calibration dataset $S$ by $1-\delta$. This guarantee is also known as training-conditional conformal prediction.
> > > >
> > > > In Figure 11, LWCP (the algorithm proposed by Sadinle et al.) provides a traditional coverage guarantee. This algorithm is straightforward to adapt to the PAC setting; basically, you apply PAC conformal prediction label-wise (instead of applying traditional conformal prediction labelwise). In Figure 11, PS-LW is this adaptation of Sadinle et al. to the PAC setting. LWCP does not satisfy the PAC coverage guarantee. In contrast, PS-LW satisfies the PAC coverage guarantee, but is overly conservative compared to our algorithm (i.e., it produces larger prediction sets on average).
> > > >
> > > > Please let us know if this addresses your concerns!

---

> > > > > ### Comment · Reviewer_onAF · 2023-11-22
> > > > >
> > > > > Thank you for the clarification. I have now raised my score.

---

### Official Review · Reviewer_mNQN · 2023-11-01

**Soundness:** 3 good
**Presentation:** 3 good
**Contribution:** 2 fair
**Rating:** 6
**Confidence:** 3

**Summary:**

This paper proposes a new conformalized procedure to construct prediction sets with PAC guarantees in the presence of label shifts. The issue of prediction sets without PAC guarantees is an interesting one, which the paper does well to highlight. The proposed method essentially tunes the parameter of the conformity score function by importance weights. The authors also provide a theory to show that the prediction sets generated by the proposed method satisfy the PAC guarantee. Empirical experiments corroborate the merits of the proposed method.

**Strengths:**

1. This paper studies an interesting problem, i.e., uncertainty sets under label shift.

2. This work is well organized, and the proposed method is well-described. I believe readers can easily get the core idea.

3. The datasets used to evaluate the algorithm are well-connected with applications, and the corresponding results demonstrate the PAC guarantee of the proposed method.

**Weaknesses:**

1. Theorem 3.2 lacks a detailed proof procedure, although the authors provide an interesting discussion on the confusion matrix in section 3.3. Please let me know where the proof is if I missed.

2. All experiments are conducted on small-scale datasets where the number of classes is small, but it is always desired to include large-scale experiments. Can you share any experimental results (e.g., on CIFAR100) compared with other methods?

3. The proposed method primarily builds upon a combination of existing methods (i.e., Clopper-Pearson intervals [1], Gaussian elimination [2]) and it doesn't present significant theoretical novelty.

I am willing to improve my score, if the authors can well address these concerns.

[1] Charles J Clopper and Egon S Pearson. The use of confidence or fiducial limits illustrated in the case of the binomial. Biometrika, 26(4):404–413, 1934.

[2] Gene H Golub and Charles F Van Loan. Matrix computations. JHU press, 2013.

**Questions:**

1. By the definition of $T_N(S_m,V,w^*,b)$ , the rejection sampling accept samples with a $1-m_{y_i}/b$ probability. But, a common operation in the rejection sampling [R1, R2 ] is to accept samples with a $m_{y_i}/b$ probability, i.e., $V_i\leq m_{y_i}/b$. This confuses me, so could you elaborate more about the reason for this difference?

2. In Figure 3, I have seen the mean error rates of PS-C and PS-W are smaller than other methods that consider label shift, such as WCP. Thus, can we directly reduce the value of $\epsilon$ to get the PAC guarantee on WCP, as shown in Corollary 1 of [R3]? And, what is the difference between Corollary 1 of [R3] and the optimization problem in Eq 16? (I think they have the same mathematical meaning.)

[R1] Park, Sangdon, et al., PAC Prediction Sets Under Covariate Shift, ICLR 2022

[R2] Pagnoni, Artidoro, Stefan Gramatovici, and Samuel Liu. "Pac learning guarantees under covariate shift." arXiv preprint arXiv:1812.06393 (2018).

[R3] Vovk, Vladimir. "Conditional validity of inductive conformal predictors." Asian conference on machine learning. PMLR, 2012.

---

> ### Author Response · Authors · 2023-11-18
>
> 1. Theorem 3.2 lacks a detailed proof procedure. Please let me know where the proof is if I missed.
>
>     A: In Section 3.3 above Theorem 3.2, we provided a single-line proof to avoid replicating proofs from prior works. We provide a sketch of the proof below and also in the revised paper.
>
>     First, our PAC guarantee in Theorem 3.2 follows from Equation 5, assuming the given confidence intervals $W_i=[\underline{w}_i,\bar{w}_i]$ for each importance weight $w_i^*$ is valid---i.e., $w_i^*\in W_i$. This guarantee follows by Theorem 4 in Park et al. (2021). Roughly speaking, if $w_i^*$ is known, then the algorithm uses a standard rejection sampling procedure based on $w_i^*$ to convert $S_m$ into a set of i.i.d. samples from $Q$. Then, the PAC guarantee follows by standard conformal prediction arguments, e.g., Park et al. (2020). When $w_i^*$ is not known, Park et al. (2021) takes the worst case over all $w_i^*\in W_i$; they use a reparameterization trick to do so in a computationally efficient way.
>
>     Our key contribution compared to prior work is showing that the confidence interval $W_i$ that we compute is a valid confidence interval---i.e., $w_i^*\in W_i$ with high probability. The proof of this fact proceeds in two steps. First, our algorithm constructs the Clopper-Pearson intervals for all $c_{ij}$ and $q_k$. Thus, by a union bound, all of these intervals are valid with high probability, which yields Equation 8. Second, our algorithm performs Gaussian elimination on these intervals; at each step of Gaussian elimination, the output it constructs is conservative with respect to the input interval---i.e., if the true value at one step of the computation is contained in the input interval, then the true value at the next step of the computation is contained in the output interval. We prove this fact by induction. As a consequence, as long as the input intervals are valid (which holds with high probability according to Equation 8), then the output intervals are also valid. This completes our proof.
>
>     We've included this in the revision.
>
> 2. Large-scale experiments on CIFAR-100.
>
>     A: We present the CIFAR-100 results in Section I.6 of the revision. The score function is a pretrained ViT model. Results show that both PS-C and PS-W still attain the desired coverage guarantee, while the remaining approaches do not. Furthermore, PS-W outperforms PS-C in terms of average prediction set size.
>
>     In this case, both PS-C and PS-W are quite conservative, due to two reasons. First, the Clopper-Pearson intervals can be conservative when $\delta$ is very small, and we need to divide $\delta$ by $K(K+1)+1$ since we need to take a union bound over $K(K+1)+1$ events in Equation 8. Second, although our Gaussian ``interval'' elimination algorithm makes the uncertainty propagation tractable (specifically in polynomial time $O(K^3)$), the resulting confidence intervals $(\underline{w}, \bar{w})$ may be conservative, thereby amplifying the first issue. With more calibration data, our approach would be less conservative.
>
> 3. The proposed method primarily builds upon a combination of existing methods (i.e., Clopper-Pearson intervals, Gaussian elimination) and it doesn't present significant theoretical novelty.
>
>     A: We are addressing rigorous PAC uncertainty quantification for the label shift problem, which is a challenging unsolved problem. We provide a solution that works both in practice and has strong theoretical guarantees. We believe this is an important contribution, given that uncertainty quantification is especially valuable when the underlying model may fail, and label shift is an important cause of failure.
>
>     Our main methodological innovation is a new algorithm to propagate uncertainty through Gaussian elimination to construct confidence intervals for the importance weights. We believe that this is a significant contribution that may have broader applications.
>
> 4. The definition of $T_N(S_m, V, w^*,b)$ the rejection sampling accept samples with a probability $1-m_{y_i}/b$. But commonly accept with $m_{y_i}/b$ probability, i.e., $V_i \leq m_{y_i}/b$.
>
>     A: Sorry for the typo, it should be $V_i \leq m_{y_i}/b$ as you suggest.
>
> 5. What is the difference between Corollary 1 of [R3] and the optimization problem in Eq 16?
>
>     A: [R3] is indeed equivalent to the PAC prediction sets method under the i.i.d. assumption [1]. Note that we have cited [R3] throughout as Vovk (2012). For instance, in Appendix A, in the Conformal Prediction paragraph, we have cited [R3]. Similarly, immediately after equation (16), we have noted that ``the approach is equivalent to the method from Vovk (2012)''. However, we use the notations from [1].
>
>     [1] Sangdon Park, Osbert Bastani, Nikolai Matni, and Insup Lee. Pac confidence sets for deep neural networks via calibrated prediction. In International Conference on Learning Representations, 2020.

---

> ### Author Response · Authors · 2023-11-18
>
> 6. Can we directly reduce the value of $\epsilon$ to get the PAC guarantee on WCP, as shown in Corollary 1 of [R3]?
>
>     A: We assume that the reviewer is referring to Propositions 2a \& 2b in [R3] (https://proceedings.mlr.press/v25/vovk12/vovk12.pdf), which establish a way to obtain PAC predcition sets by reducing $\epsilon$ by $\sqrt{\frac{\log(1/\delta)}{2n}}$ (as stated just after the statement of Proposition 2a). The strategy for directly reducing $\epsilon$ proposed in [R3] cannot be directly applied WCP, since the binomial probability distribution in equation 7 in [R3] becomes a sum over $\text{Bernoulli}(p)$ random variables with different parameter values $p$, and this sum is much harder to bound. Park et al. (2021) addresses this issue by instead using rejection sampling.
>
>     Even if achieving a similar reduction is possible, this strategy assumes the true importance weights are known. In other words, this technique would suffer from the same issue as PS-R, which does not satisfy the PAC guarantee due to errors in the importance weight estimates. Incorporating importance weight intervals into equation 7 in [R3] makes the problem even more challenging.
>
>     Finally, we note that even if such a strategy is possible, it would still need to be combined with our interval Gaussian elimination algorithm to obtain PAC coverage guarantees.

---

> > ### Comment · Reviewer_mNQN · 2023-11-21
> > **Raising the score**
> >
> > Thanks for the responses with additional discussion! With further clarifications and the discussion on the related works, the previous concerns are well-addressed as follows:
> >
> >     1. Complete an additional experiment on CIFAR-100;
> >
> >     2. Revise some ambiguous statements;
> >
> >     3. The necessity of importance weight estimates;
> >
> > Overall, regarding the quality and the technical novelty of this work, the reviewer decided to raise the score accordingly after going through all the previous details again. Hope the previous comments can better improve the draft.

---

### Official Review · Reviewer_y3Aa · 2023-11-04

**Soundness:** 2 fair
**Presentation:** 3 good
**Contribution:** 2 fair
**Rating:** 6
**Confidence:** 3

**Summary:**

The paper studies the problem of producing calibrated prediction sets under label shift. The authors propose a method with formal PAC guarantees that is based on propagating uncertainty via a Gaussian elimination algorithm.

**Strengths:**

- The paper is well-written and very clear
- It studies an important practical problem: calibrated uncertainty estimation
- It introduces a novel approach with formal guarantees

**Weaknesses:**

- I am not entirely convinced by the empirical results. In many tables, certain models get better prediction set error, but the prediction set size can be worse, and vice-versa. It would be better to compare the performance of all methods using one metric. A proper scoring rule is a standard method to evaluate the quality of predictive forecasts.

**Questions:**

- Can performance be summarized by a single metric and what happens to performance relative to baselines in that case?

---

> ### Author Response · Authors · 2023-11-18
>
> 1. Can performance be summarized by a single metric and what happens to performance relative to baselines in that case?
>
>    A: Our strategy for evaluating prediction sets is based on prior work. In particular, our goal is to (i) achieve the smallest average prediction set size, while (ii) satisfying the desired coverage guarantee (i.e., our error falls below the threshold denoted by the dashed line). In principle, we could combine these into a single performance metric, which is the average prediction set size if the coverage guarantee is satisfied, and infinity otherwise. However, we believe that reporting these metrics separately is more interpretable.
>
>     According to this goal, among all the methods, only PS-C and PS-W always satisfy the coverage guarantee. PS-C tends to be very conservative; thus, as long as we have sufficient data, PS-W outperforms PS-C. In addition, note that PS-C also relies on our modified Gaussian elimination methods to estimate $\max_{k\in[K]}\overline{w}_k$, so it can be considered an ablation of our technique.
>
>     Finally, we note that proper scoring rules are not applicable in our setting, because we do not predict probabilities. Instead, we are quantifying uncertainty by predicting sets of values.

---

> > ### Comment · Reviewer_y3Aa · 2023-12-05
> > **Acknowledgement**
> >
> > Thank you, I acknowledge the response. I maintain my current score for this paper.

---

### Author Response · Authors · 2023-11-18

We appreciate the reviewers for their detailed and thoughtful feedback. We are encouraged by their recognition of the significance of providing a formal PAC guarantee under the challenging label shift problem (y3Aa, mNQN, 2Pyj, tiZb), the novelty and notability in our uncertainty propagation method (y3Aa, 2Pyj, tiZb), and the strength and comprehensiveness of our presentation (y3Aa, mNQN, onAF, 2Pyj, tiZb). We are glad that the effectiveness and practical value (y3Aa, tiZb) of our solution is recognized by the enriched experiments across several real-world datasets (mNQN, tiZb). Our setups are considered clear with appropriate comparisons to contemporary literature (mNQN).

We've addressed the reviewer comments below and incorporate all feedback in the revision.

---

### Meta-Review · Area_Chair_ujsx · 2023-12-10

**Metareview:**

This paper develops a method to construct calibrated prediction intervals under label shift. The proposed method outputs intervals that are robust to label shifts and benefits from PAC guarantees. The proposed approach is sound and validated analytically and experimentally.

**Strengths**

- Important problem
- Writing is clear and concise
- Sound technical approach

**Weaknesses**

- Originality: The contributions appear incremental, given prior work on PAC Prediction Intervals under Covariate Shift by [Park et al.](https://openreview.net/forum?id=DhP9L8vIyLc) from ICLR 2022.

- Lack of Demonstrations and Use Cases: The experiments primarily highlight how the proposed can be shown to obey the PAC Guarantee while others violate it. We are left to ask -- why do these violations matter? What is the impact of ignoring them? I would recommend the authors to address these questions through demonstrations where they use individual-level uncertainty estimates in practice (e.g., selective classification or active learning). This would allow the authors to highlight the value of coverage in a way that is more convinging.

**Justification For Why Not Higher Score:**

This paper received lukewarm reviews from most reviewers. The contributions appear incremental, given prior work on PAC Prediction Intervals under Covariate Shift by [Park et al.](https://openreview.net/forum?id=DhP9L8vIyLc) at ICLR 2022. The potential impact is limited as the work is not developed in a way that would make it interesting to a broader audience. In this case, there is substantial room for improvement through experiments or demonstrations that would highlight the practical benefits of ensuring the PAC Guarantee.

**Justification For Why Not Lower Score:**

The paper received positive reviews across the board. The proposed work includes a sound solution to an important class of problems.

---

### Decision · Program_Chairs · 2024-01-16

Accept (poster)